# Improved Bald Eagle Search Optimization Algorithm for the Inverse Kinematics of Robotic Manipulators

**DOI:** 10.3390/biomimetics9100627

**Published:** 2024-10-15

**Authors:** Guojun Zhao, Bo Tao, Du Jiang, Juntong Yun, Hanwen Fan

**Affiliations:** 1Key Laboratory of Metallurgical Equipment and Control Technology of Ministry of Education, Wuhan University of Science and Technology, Wuhan 430081, China; jiangdu@wust.edu.cn (D.J.); yunjuntong@wust.edu.cn (J.Y.);; 2Precision Manufacturing Research Institute, Wuhan University of Science and Technology, Wuhan 430081, China; taoboq@wust.edu.cn

**Keywords:** inverse kinematics, robotic manipulators, bald eagle search optimization algorithm, Lévy flight strategy, engineering optimization problem

## Abstract

The inverse kinematics of robotic manipulators involves determining an appropriate joint configuration to achieve a specified end-effector position. This problem is challenging because the inverse kinematics of manipulators are highly nonlinear and complexly coupled. To address this challenge, the bald eagle search optimization algorithm is introduced. This algorithm combines the advantages of evolutionary and swarm techniques, making it more effective at solving nonlinear problems and improving search efficiency. Due to the tendency of the algorithm to fall into local optima, the Lévy flight strategy is introduced to enhance its performance. This strategy adopts a heavy-tailed distribution to generate long-distance jumps, thereby preventing the algorithm from becoming trapped in local optima and enhancing its global search efficiency. The experiments first evaluated the accuracy and robustness of the proposed algorithm based on the inverse kinematics problem of manipulators, achieving a solution accuracy of up to $10^{-18}$ m. Subsequently, the proposed algorithm was compared with other algorithms using the CEC2017 test functions. The results showed that the improved algorithm significantly outperformed the original in accuracy, convergence speed, and stability. Specifically, it achieved over 70% improvement in both standard deviation and mean for several test functions, demonstrating the effectiveness of the Lévy flight strategy in enhancing global search capabilities. Furthermore, the practicality of the proposed algorithm was verified through two real engineering optimization problems.

## 1. Introduction

The inverse kinematics (IK) of robotic manipulators is of great research importance for robotics applications [1,2,3]. It forms the foundation of research on robotic tasks such as visual guidance, object grasping, trajectory tracking, and motion control [4]. The IK problem is highly nonlinear and involves complex coupling [5]. Additionally, different manipulators have unique structures and workspaces, resulting in diversity in how the IK problem manifests across various scenarios [6]. To address these challenges, it is necessary to develop a generalized IK solution method that can be applied to various manipulators while effectively handling singularities and workspace constraints [7]. This will help advance the development of robotic technology, making it more flexible and applicable across a wide range of application areas.

Traditional closed-form methods, such as geometric and algebraic approaches, offer precise solutions for inverse kinematics [8,9]. However, as the degrees of freedom (DOF) of a manipulator increase, implementing closed-form methods becomes increasingly challenging and less adaptable to different scenarios [10]. Numerical methods are widely used for their capability to provide approximations and adaptability to various manipulator structures [11,12,13]. However, a major limitation of these methods is their susceptibility to singularities. Since they determine joint configurations through the inverse Jacobian matrix, some configurations may lead to a singular Jacobian matrix, causing unrealistic joint velocities. Considering the limitations of both closed-form and numerical methods [14], it is recommended to employ metaheuristic algorithms to address the IK problem for multiple manipulators, thereby avoiding singularities [15].

Metaheuristic algorithms have gained significant attention due to their wide application in practical engineering problems. For example, the tree seed algorithm (TSA) [16] has been applied to parameter estimation in photovoltaic models, the improved tree seed algorithm (I-TSA) [17] has been used for optimizing the parameters of Butterworth and Bessel filters, and the sine cosine algorithm (SCA) [18] has been applied to optimize the cross section of an aircraft wing.

Metaheuristic algorithms can be categorized into two main types: swarm intelligence optimization algorithms and evolutionary algorithms [19]. Swarm intelligence optimization algorithms enhance search efficiency by updating individual positions based on collective movement [20]. The particle swarm optimization algorithm (PSO) has commonly been employed to solve the IK problem in various manipulator structures [21,22,23,24]. Additionally, the firefly algorithm (FA) and its variants have also been widely used [25,26,27]. Moreover, other swarm intelligence optimization algorithms, such as the golden eagle optimizer algorithm (GEO) [28], the improved carnivorous plant algorithm (I-CPA) [29], and the multi-strategy-based tree seed algorithm (MS-TSA) [30] have shown promising performance in specific engineering application scenarios, such as solar photovoltaic parameter estimation, demonstrating enhanced convergence speed and solution accuracy.

Evolutionary algorithms, known for their effectiveness in solving non-linear problems, have also been applied to the IK problem of manipulators [31]. Notably, the genetic algorithm (GA) and differential evolution algorithm (DE) have gained widespread use for solving the IK problem in various types of manipulators [32,33,34,35]. The main metaheuristic algorithms for solving inverse kinematics include PSO, FA, GA, DE, and their improved versions. Among these algorithms, PSO demonstrates the best performance in terms of convergence accuracy. Therefore, swarm intelligence optimization algorithms that are suitable for the inverse kinematics of manipulators offer better performance and have gained wider recognition.

Nonetheless, despite the success of metaheuristic algorithms, challenges such as premature convergence, lack of solution diversity, and high computational costs remain prevalent. Therefore, further research is warranted to enhance these algorithms or to develop hybrid approaches that can effectively address the challenges posed by engineering application problems, as exemplified by the eagle strategy [36].

The bald eagle search optimization algorithm (BES) [37] combines the advantages of evolutionary and swarm techniques, making it more effective at solving nonlinear problems and improving search efficiency. For specific applications of BES, Fathy [38] used BES to solve the optimization problem of the maximum power point in a power generation system, while Eid [39] used BES to solve the optimization problem of optimal power distribution. For multi-objective optimization tasks, Yang [40] improved BES and proposed the multi-objective bald eagle search algorithm (MO-BES). By introducing an archiving mechanism and an elite selection strategy, the capability of BES in solving multi-objective tasks has been enhanced.

Based on the above study, it is clear that BES has a wide range of applications. However, BES still suffers from the problem of falling into local optima when dealing with complex functions. To address this issue, this paper proposes an improved bald eagle search optimization algorithm (I-BES). By introducing the Lévy flight strategy [41] in the search phase of the BES, the algorithm avoids falling into local optima too early, thereby improving its global search efficiency. In this work, the I-BES is applied to solve the inverse kinematics problem of manipulators. The contributions of this paper are as follows:In I-BES, the Lévy flight strategy is introduced to improve the global search efficiency of BES;An inverse kinematic solution method for n-DOF manipulators based on I-BES is proposed;The performance of I-BES is evaluated using the CEC 2017 test suite;The application scope of I-BES is broadened by applying it to two engineering design problems.

The rest of this paper is organized as follows: Section 2 introduces the fundamental concepts of forward and inverse kinematics for robotic manipulators. Section 3 provides a description of BES and its improvement points. Section 4 presents the inverse kinematics solution method based on I-BES. Section 5 validates I-BES using the inverse kinematics problem, the CEC2017 test suite, and two other engineering design problems. Section 6 summarizes the findings and contributions of this paper.

## 2. Robotic Manipulator Kinematics

Robotic manipulators are composed of interconnected links and joints, forming open kinematic chains [6]. The terminal element of this chain is known as the end-effector. Each joint corresponds to a specific articulation, together determining the joint configuration $q={\left[q_{1}\;q_{2}\;q_{3}\;\ldots \;q_{n}\right]}^{T}$, where *n* represents the total DOF of the manipulator, as illustrated in Figure 1.

Figure 1 shows the schematic diagram of the kinematic chain for a serial robotic manipulator. (*x*_0_, *y*_0_, *z*_0_) represents the 3D coordinates of the base in Cartesian space, while (*x*_*n*_, *y*_*n*_, *z*_*n*_) represents the 3D coordinates of the end-effector in Cartesian space. ^*j*−1^**T**_*j*_ represents the homogeneous transformation matrix from link *j* to link *j* − 1, and *n* represents the total number of degrees of freedom.

There are two primary categories of joints: revolute and prismatic. The joint variable can be defined as follows: (1)$$q_{j}=\left\{\begin{array}{cc} \theta_{j} & \mathrm{if}\mathrm{joint}j\mathrm{is}a\mathrm{revolute}\\ d_{j} & \mathrm{if}\mathrm{joint}j\mathrm{is}a\mathrm{prismatic} \end{array}\right.$$
where *q*_*j*_ represents the *j*-th joint variable, while θ_*j*_ and *d*_*j*_ correspond to the *j*-th rotation angle and displacement, respectively. Forward kinematics determine the position and orientation of the end-effector from a given joint configuration. It is generally a straightforward problem with a guaranteed solution. The forward kinematic expression is given as follows:
(2)Tn0(q)=T10(q1)T21(q2)…Tnn−1(qn)Tn0(q)=∏j=1nTnj−1(qj)
where Tn0 represents the pose transformation matrix of the end-effector relative to the base coordinate system. The joint vector q is defined as [*q*_1_, *q*_2_, …, *q*_7_]. Tnj−1 represents the homogeneous transformation matrix from link *j* to link *j* − 1, where *q*_*j*_ represents the *j*-th joint angle, and *n* is the total number of degrees of freedom.

Utilizing the standard Denavit–Hartenberg (DH) modeling method [42], the homogeneous transformation matrix Tnj−1 can be expressed as follows: (3)Tnj−1=cθj−sθjcαjsθjsαjajcθjsθjcθjcαj−cθjsαjajsθj0sαjcαjdj0001
where the symbols *s* and *c* represent the sine and cosine functions, respectively, and θ_*j*_ and α_*j*_ represent the *j*-th joint angle and the *j*-th twist angle, respectively. *a*_*j*_ and *d*_*j*_ represent the link length and offset of the *j*-th link, respectively. For revolute joints, θ_*j*_ is considered as the joint variable, while for prismatic joints, *d*_*j*_ is considered as the joint variable. The correspondence of link parameters between the neighboring links of the manipulator is shown in Figure 2.

Figure 2 shows the correspondence of link parameters between the (i−1)-th link and the *i*-th link of the manipulator. where *a*_*i*_ represents the distance along the *x*_*i*_ axis from the *z*_*i*−1_ axis to *z*_*i*_ axis. *d*_*i*_ represents the distance along the *z*_*i*−1_ axis from the *x*_*i*−1_ axis to the *x*_*i*_ axis. α_*i*_ represents the rotation angle around the *x*_*i*_ axis from the *z*_*i*−1_ axis to the *z*_*i*_ axis. θ_*i*_ represents the rotation angle around the *z*_*i*−1_ axis from the *x*_*i*−1_ axis to the *x*_*i*_ axis.

The homogeneous transformation matrix Tjj−1 can also be expressed as follows: (4)Tjj−1=nxoxaxpxnyoyaypynzozazpz0001=Rp01
where $n_{x},n_{y},n_{z}$ represent the projections of the new x-axis onto the *x*, *y*, and *z* axes of the original coordinate system after rotation, respectively. Similarly, $o_{x},o_{y},o_{z}$ and $a_{x},a_{y},a_{z}$ represent the projections of the new y-axis and z-axis onto the axes of the original coordinate system. The terms $p_{x},p_{y},p_{z}$ indicate translations along the *x*, *y*, and *z* directions. ***R*** is a 3 ×  3 orthogonal matrix describing the change in orientation of the manipulator, while ***p*** is a 3 × 1 position vector indicating the change in position.

Solving the inverse kinematics using metaheuristic algorithms usually involves minimizing the error between the desired end-effector position and the actual one. Figure 3. shows the error function for a 2DOF planar manipulator. This manipulator consists of two links, each with a length of 0.5 m. The error function $\sqrt{(xd-x)2+(yd-y)2}$ is composed of the desired position (*x*_*d*_ = 0.5 m, *y*_*d*_ = 0.5 m), and the actual position (*x*, *y*) is obtained via a specified joint configuration (θ_1_, θ_2_). Minimizing the error function using the metaheuristic algorithm can obtain two sets of joint configurations, (1.5708 rad, −1.5708 rad) and (0 rad, 1.5708 rad), respectively.

Figure 3 shows a contour plot illustrating the regions where the objective function maintains the same value with respect to $\theta_{1}$ and $\theta_{2}$. Different colors indicate varying values of the objective function, and the color gradient reflects its variation trend. Denser contour lines signify more rapid changes in the objective function, while smoother areas suggest minimal variation.

In this work, I-BES is employed to determine the optimal joint configuration for a specified end-effector position. The proposed method effectively addresses the limitations of traditional approaches. For a comprehensive understanding of both forward and inverse kinematics, detailed descriptions can be found in references [6,8].

## 3. The Bald Eagle Search Optimization Algorithm and Its Improvement

### 3.1. Bald Eagle Search Optimization Algorithm

BES combines the strengths of both evolutionary and swarm techniques [37]. On the one hand, BES leverages swarm techniques to enhance search efficiency by updating individual positions based on collective movement. On the other hand, BES utilizes evolutionary techniques to ensure frequent exploration of the search space and avoid convergence to local optima. Compared to other swarm intelligence optimization algorithms, BES offers greater advantages in solving nonlinear problems.

The bald eagle search optimization algorithm emulates the prey-hunting process of bald eagles in nature, consisting of three phases: the selection phase, the search phase, and the swoop phase (Figure 4). The selection phase, also known as the relay phase, primarily aims to gather all search points from the starting position to the optimal point. The search phase is analogous to the evolutionary phase, focusing on concentrating preferred search points towards the centroid. The swoop phase resembles swarm intelligence techniques and aims to move the search point from the local centroid to the optimal point. This third phase mitigates the limitations of the first two by providing guided exploration over a broader area.

In the selection phase, bald eagles choose the search space of prey. The behavior is expressed mathematically by Equation (Equation 5).
(5)$$x_{i,\mathrm{new}}=x_{\mathrm{best}}+\alpha r\left(x_{\mathrm{mean}}-x_{i}\right)$$
$$x_{\mathrm{mean}}=\frac{1}{N}\sum_{i=1}^{N}x_{i}$$
where $x_{i,\mathrm{new}}$ represents the position of the *i*-th bald eagle after selecting the search space. $x_{\mathrm{best}}$ represents the best position of the bald eagles in the current search space. α is the parameter that controls the change in position and takes values in the range of [1.5, 2]. *r* is a random number with values in the range of [0, 1]. $x_{\mathrm{mean}}$ represents the average position of the bald eagles. $x_{i}$ represents the position of the *i*-th bald eagle. *N* represents the population size.

In the search phase, the bald eagles search for the chosen space in a spiral motion. This behavior is expressed mathematically by Equation (Equation 6).
(6)$$x_{i,\mathrm{new}}=x_{i}+n_{1}\left(i\right)\left(x_{i}-x_{i+1}\right)+m_{1}\left(i\right)\left(x_{i}-x_{\mathrm{mean}}\right)$$
$$m_{1}\left(i\right)=\frac{{mr}_{1}(i)}{max(\left|{mr}_{1}\right|)},n_{1}\left(i\right)=\frac{{nr}_{1}(i)}{max(\left|{mr}_{1}\right|)}$$
$${mr}_{1}(i)=r_{1}(i)sin(\lambda_{1}(i)),{nr}_{1}(i)=r_{1}(i)cos(\lambda_{1}(i))$$
$$\lambda_{1}(i)=arand,r_{1}(i)=\lambda_{1}(i)+Rrand$$
where *a* represents the angle between the bald eagle and the target prey during the spiral motion, which takes values in the range of [5, 10]. *R* represents the number of search cycles and takes values in the range of [0.5, 2]. *rand* represents a random number in the interval range of [0, 1]. It should be noted that, to ensure that ***x***_*i*+1_ is meaningful, *i* takes values in the range of [1, *N* − 1] during the search phase. In the selection and swoop phases, *i* takes values in the range of [1, *N*].

In the swoop phase, the bald eagles move quickly from the optimal search space to the target prey. This behavior is expressed mathematically by Equation (Equation 7).
(7)$$x_{i,\mathrm{new}}=kx_{\mathrm{best}}+m_{2}\left(i\right)A+n_{2}\left(i\right)B$$
$$A=(x_{i}-c_{1}x_{\mathrm{mean}}),B=(x_{i}-c_{2}x_{\mathrm{best}})$$
$$m_{2}\left(i\right)=\frac{{mr}_{2}(i)}{max(\left|{mr}_{2}\right|)},n_{2}\left(i\right)=\frac{{nr}_{2}(i)}{max(\left|{nr}_{2}\right|)}$$
$${mr}_{2}(i)=r_{2}(i)sinh(\lambda_{2}(i)),{nr}_{2}(i)=r_{2}(i)cosh(\lambda_{2}(i))$$
$$\lambda_{2}(i)=ak,r_{2}(i)=\lambda_{2}(i)$$
where *k* is a random number in the range of [0.5, 2]. Both *c*_1_ and *c*_2_ represent learning factors, with values in the range of [1, 2].

### 3.2. Improved Bald Eagle Search Optimization Algorithm

In BES, the search phase employs a stochastic search strategy based on a uniform distribution, which makes the algorithm prone to falling into local optima in complex, high-dimensional optimization problems and lacking sufficient exploration capabilities. To enhance the global search performance of the algorithm while maintaining a balance in the local exploitation phase, I-BES introduces the Lévy flight strategy [41]. Lévy flight is a stochastic wandering pattern characterized by a heavy-tailed distribution, which is capable of generating long-distance jumps to avoid falling into local optima and improve the jumping ability and global search efficiency of BES.

Lévy flight is a stochastic search pattern based on the Lévy distribution, where the step size also follows the Lévy distribution:
(8)$$L=0.01\cdot \frac{\sigma \cdot \mathrm{randn}(1,dim)}{{\left|\mathrm{randn}\left(1,dim\right)\right|}^{\frac{1}{\beta }}}$$
$$\sigma ={\left(\frac{\Gamma \left(1+\beta \right)\cdot \sin\left(\frac{\pi \beta }{2}\right)}{\Gamma \left(\frac{1+\beta }{2}\right)\cdot \beta \cdot 2^{\frac{\beta -1}{2}}}\right)}^{\frac{1}{\beta }}$$
where *L* represents the Lévy flight step, and *dim* represents the dimension of the problem. The parameter β controls the heavy-tailedness of the distribution and is usually set to around 1.5. The Lévy distribution is characterized by its ability to generate a small number of large steps and a large number of small steps, which achieves a dynamic balance between local search and large-scale exploration. Compared with traditional uniform and normal distributions, the Lévy distribution is more effective in avoiding premature convergence in the search process and in enhancing the ability of the algorithm to perform long-distance jumps.

In the search phase of I-BES, this paper introduces the Lévy flight strategy to replace the original step size generation method in BES. By generating the step length *L* according to the Lévy distribution during each update of the algorithm, the solution is updated as follows:(9)$$x_{i,\mathrm{new}}=x_{i}+L\left(x_{i}-x_{\mathrm{best}}\right)$$

The step length *L* is generated based on the Lévy distribution, which can produce large jumps to help the algorithm quickly escape from local optima and improve the overall search efficiency.

## 4. Algorithm Adjustment for Inverse Kinematics Problem

To solve the inverse kinematics of robotic manipulators, it is essential to define the pose of the end-effector, which can be represented as follows:(10)$$T_{d}=\left[\begin{array}{cc} R_{d} & p_{d}\\ 0 & 1 \end{array}\right]$$
where $T_{d}$ represents the desired pose matrix of the end-effector. $R_{d}$ and $p_{d}$ represent the desired orientation matrix and position vector of the end-effector, respectively.

In I-BES, the *i*-th bald eagle represents the candidate joint configuration $q_{i}\in R^{n}$, where *n* represents the total DOF of the manipulator, corresponding to the dimensionality of the problem. The forward kinematics formula given in Equation (Equation 2) can be derived based on the joint configuration $q_{i}$. The position vector $p_{i}$ can then be calculated using Equation (Equation 4).

The objective function is designed to compare the error between the desired position $p_{d}$ and the candidate position $p_{i}$ of the end-effector (Figure 5). Its mathematical expression is shown as follows:(11)$$f\left(q_{i}\right)={∥p_{d}-p_{i}\left(q_{i}\right)∥}_{2}$$
where *f* represents the objective function. $q_{i}$ represents the joint configuration corresponding to the *i*-th bald eagle. $p_{d}$ and $p_{i}$ represent the desired position and actual position of the end-effector, respectively. ∥·∥_2_ represents the Euclidean norm.

In the actual application of robotic manipulators, the focus is more on whether the manipulators can reach the specified position. Considering this, it is advisable to use the objective function described in Equation (Equation 11).

In this work, a method is proposed to address the inverse kinematics of robotic manipulators by formulating them as a constrained optimization problem, defined as follows:(12)$$minf\left(q_{i}\right),subject\;to\;q_{l}\leq q_{i}\leq q_{u}$$
where $q_{l}$ and $q_{u}$ are the lower and upper joint constraints of the manipulator, respectively.

There are several advantages to transforming the inverse kinematics problem of a manipulator into an optimization problem by minimizing the objective function. Firstly, it ensures that the obtained solution is optimal according to the defined criteria, effectively minimizing errors or deviations. Secondly, this approach provides flexibility in addressing complex, nonlinear problems by reformulating them as an objective function that can be optimized efficiently. Furthermore, focusing on minimizing the objective function allows for the incorporation of constraints into the model, ensuring that the solution complies with real-world limitations while achieving the most accurate or optimal result.

The BES was originally designed for operation in unconstrained continuous spaces. To introduce constraints into the I-BES optimization process, the following scheme is recommended.
(13)$$q_{\mathrm{ij}}=\left\{\begin{array}{cc} q_{\mathrm{ij}} & \mathrm{if}\;q_{\mathrm{lj}}\leq q_{\mathrm{ij}}\leq q_{\mathrm{uj}}\\ q_{\mathrm{ij}}+k\left(q_{\mathrm{uj}}-q_{\mathrm{lj}}\right) & \mathrm{Otherwise}\; \end{array}\right.$$
where *q*_*ij*_ represents the *j*-th joint angle in the joint configuration corresponding to the *i*-th bald eagle. *k* is a random number within the range of [0, 1]. *q*_*lj*_ and *q*_*uj*_ represent the minimum and maximum values of the joint angle in the joint configuration corresponding to the *i*-th bald eagle, respectively. For each bald eagle, the different joint configurations are defined as follows:(14)$$\begin{array}{c} q_{i}={\left[q_{i1}\;q_{i2}\;q_{i3}\;\ldots \;q_{\mathrm{in}}\right]}^{T}\\ q_{l}={\left[q_{l1}\;q_{l2}\;q_{l3}\;\ldots \;q_{ln}\right]}^{T}\\ q_{u}={\left[q_{u1}\;q_{u2}\;q_{u3}\;\ldots \;q_{un}\right]}^{T} \end{array}$$
where *n* represents the DOF of the manipulator. The proposed scheme determines whether the candidate joint value *q*_*ij*_ is a feasible or infeasible joint solution. Feasible solutions do not require any adjustments. Conversely, infeasible solutions arise when *q*_*ij*_ falls outside the joint limit boundaries. In such cases, these values are recalculated to satisfy the joint constraints.

The pseudo-code and flowchart for solving the inverse kinematics of robotic manipulators via I-BES are shown in Algorithm 1 and Figure 6, respectively.
**Algorithm 1** Solving the inverse kinematics of robotic manipulators based on I-BES**Require:** 
$c_{1},c_{2},R,\alpha ,a,q_{l},q_{u}$**Ensure:** $f(q_{i})=∥p_{d}-p_{i}(q_{i})∥$1:$initialize\;q_{i}\;where\;i=1,2,\ldots ,N$2:$calculate\;the\;fitness\;value:\;f(q_{i})$3:**while**thenumberofiterations<thegivenvalue**do**4:    **for** i=1:N **do**5:        $q_{i,\mathrm{new}}=q_{\mathrm{best}}+$$r(q_{\mathrm{mean}}-q_{i})$6:        checkthefeasibilityofthesolutionby Equation (Equation 13)7:        **if** $f(q_{i,\mathrm{new}})<f(q_{i})$ **then**8:           $q_{i}=q_{i,\mathrm{new}}$9:           **if** $f(q_{i,\mathrm{new}})<f(q_{\mathrm{best}})$ **then**10:               $q_{\mathrm{best}}=q_{i,\mathrm{new}}$11:           **end if**12:        **end if**13:    **end for**14:    **for** i=1:N **do**15:        $q_{i,\mathrm{new}}=q_{i}+L(q_{i}-q_{\mathrm{best}})$16:        checkthefeasibilityofthesolutionby Equation (Equation 13)17:        **if** $f(q_{i,\mathrm{new}})<f(q_{i})$ **then**18:           $q_{i}=q_{i,\mathrm{new}})$19:           **if** $f(q_{i,\mathrm{new}})<f(q_{\mathrm{best}})$ **then**20:               $q_{\mathrm{best}}=q_{i,\mathrm{new}}$21:           **end if**22:        **end if**23:    **end for**24:    **for** i=1:N **do**25:        $q_{i,\mathrm{new}}=kq_{\mathrm{best}}+m_{2}(i)A+n_{2}(i)B$26:        $where\;A=(q_{i}-c_{1}q_{\mathrm{mean}}),B=(q_{i}-c_{2}q_{\mathrm{best}})$27:        checkthefeasibilityofthesolutionby Equation (Equation 13)28:        **if** $f(q_{i,\mathrm{new}})<f(q_{i})$ **then**29:           $q_{i}=q_{i,\mathrm{new}})$30:           **if** $f(q_{i,\mathrm{new}})<f(q_{\mathrm{best}})$ **then**31:               $q_{\mathrm{best}}=q_{i,\mathrm{new}}$32:           **end if**33:        **end if**34:    **end for**35:    $preserve\;the\;optimal\;solution\;q_{\mathrm{best}}$36:**end while**

The initial population is generated using a random sampling strategy within the feasible solution space. This method ensures that the initial individuals are uniformly distributed across the search space, providing a diverse set of solutions for the optimization process. By employing this strategy, premature convergence is avoided, and the ability of the algorithm to effectively explore the entire solution space is improved.

Specifically, the initial joint angles for each individual are randomly sampled within the joint limits of the manipulator, considering both minimum and maximum allowable values. This guarantees that all initial configurations are kinematically feasible and fall within the operational workspace of the manipulator. Additionally, for each candidate solution, forward kinematics are computed to verify that the initial pose satisfies the desired reachability criteria, thereby improving the quality of the initial population.

## 5. Simulation and Experimental Results

### 5.1. Inverse Kinematics Problem of Manipulators

#### 5.1.1. Preparation Work

Both simulations and experiments were conducted to evaluate the performance of the proposed method in solving the inverse kinematics of robotic manipulators. The main objective was to evaluate the ability of the proposed method to determine the optimal joint configuration to reach a predefined end-effector position. Additionally, the robustness of the proposed method was validated by testing it on robotic manipulators with two to seven DOF.

In subsequent simulations and experiments, I-BES was compared with other metaheuristic algorithms. The parameters shared among the algorithms are the total number of iterations and the population size, both set to 500 and 150, respectively. The specific parameter configurations for each algorithm are shown in Table 1.

To effectively present the simulation results and demonstrate the accuracy of the algorithms, the following error is proposed to be measured:(15)$$\epsilon =∥p_{d}{-p\left(q\right)∥}_{2}$$
where ϵ represents the position error. $p_{d}$ and p represent the desired position and the actual position, respectively. q represents the joint configuration calculated by the algorithm.

The robotic manipulators considered for the simulations include a 2DOF planar manipulator, a 3DOF anthropomorphic manipulator, a 4DOF scara manipulator, a 5DOF kuka youbot manipulator, a 6DOF puma 560 manipulator, and a 7DOF rethink sawyer manipulator, which are commonly used as educational examples in robot kinematics [5,6,8]. The DH parameters for these manipulators are shown in Table 2, Table 3, Table 4, Table 5, Table 6 and Table 7, respectively.

For the convenience of the experiment, the joint limits of the different DOF manipulators were uniformly specified. In the simulations, the joint limits were set as follows:(16)$$\begin{array}{c} q_{\mathrm{lj}}=\left\{\begin{array}{cc} -\pi & \mathrm{if}\;joint\;j\;is\;a\;revolute\;joint\\ 0 & \mathrm{if}\;joint\;j\;is\;a\;prismatic\;joint \end{array}\right.\\ q_{\mathrm{uj}}=\left\{\begin{array}{cc} \pi & \mathrm{if}\;joint\;j\;is\;a\;revolute\;joint\\ 1.0 & \mathrm{if}\;joint\;j\;is\;a\;prismatic\;joint \end{array}\right. \end{array}$$
where $q_{\mathrm{lj}}$ and $q_{\mathrm{uj}}$ represent the lower and upper limits of the *j*-th joint, respectively.

Figure 7a–f display three-dimensional surface plots of the objective function values for six different DOF manipulators. The horizontal and vertical axes correspond to parameters $\theta_{1}$ and $\theta_{2}$, while the vertical axis indicates the objective function values for each parameter combination.

In Figure 7a–f, the extrema of the function are represented by the peaks (maxima) and valleys (minima) on the surface. These points illustrate how the interaction between joint angle pairs influences the objective function. The desired end-effector positions for the six manipulators are displayed below.

−The desired position of the end-effector for a 2DOF planar manipulator:
$$p_{d}=[0.35\;0.35\;0.00]T.$$−The desired position of the end-effector for a 3DOF anthropomorphic manipulator:
$$p_{d}=[0.30\;-0.20\;0.35]T.$$−The desired position of the end-effector for a 4DOF scara manipulator:
$$p_{d}=[0.40\;0.20\;0.10]T.$$−The desired position of the end-effector for a 5DOF kuka youbot manipulator:
$$p_{d}=[0.40\;0.20\;0.10]T.$$−The desired position of the end-effector for a 6DOF puma 560 manipulator:
$$p_{d}=[0.40\;-0.20\;0.50]T.$$−The desired position of the end-effector for a 7DOF rethink sawyer manipulator:
$$p_{d}=[0.80\;0.20\;0.60]T.$$

#### 5.1.2. Simulations

The simulations included position errors across 25 tests, variations in the objective function over 500 iterations, and variations in the optimal fitness value across 25 tests. The metaheuristic algorithms compared were BES, PSO, FA, the modified tree–seed algorithm (MTSA) [43], and the improved chaotic particle swarm optimization algorithm (CPSO-AT) [44].

The simulations were conducted using MATLAB™ (R2018a). Each algorithm was executed 25 times, and the statistical results of the position errors were visualized using box plots. Algorithms with smaller data distributions, lower result values, and fewer outliers in the box plots demonstrated superior performance. The performance of each algorithm was evaluated based on the optimal joint configuration required to reach the predefined position of the end-effector.

Figure 8, Figure 9, Figure 10, Figure 11, Figure 12 and Figure 13 show the simulation results for manipulators with 2 to 7 DOF, respectively. According to Figure 8a, Figure 9a, Figure 10a, Figure 11a, Figure 12a and Figure 13a, FA and CPSO-AT exhibit wider data distributions and higher position error values. It is worth noting that FA presents outliers in Figure 8a, Figure 9a, Figure 10a and Figure 13a. Compared to FA and CPSO-AT, I-BES, BES, PSO, and MTSA demonstrate smaller data distributions, lower result values, and fewer outliers.

According to Figure 8b, Figure 9b, Figure 10b, Figure 11b, Figure 12b, and Figure 13b, I-BES and BES outperform PSO and MTSA. Specifically, in Figure 8b, Figure 9b, Figure 10b, Figure 11b, and Figure 13b, MTSA demonstrates a wider data distribution and higher position error values. However, in Figure 12b, PSO shows a wider data distribution and higher position error values. It is also noteworthy that MTSA presents outliers in Figure 9b, Figure 10b, Figure 11b, and Figure 12b.

Table 8 presents the statistical results of position errors across 25 tests. The results indicate that the position errors for each algorithm remain consistent as the DOFs of the manipulators increase, highlighting the robustness of the metaheuristic algorithms. Among the compared algorithms, I-BES and BES demonstrate the highest positional accuracy, followed by PSO. Generally, the positional error for I-BES stabilizes between 0 m and 10^−17^ m, for BES between 0 m and 10^−16^ m, and for PSO between 10^−10^ m and 10^−7^ m. Furthermore, I-BES exhibits fewer outliers compared to BES and PSO. By comparing the average, median, minimum, and maximum values of the position errors, it is evident that I-BES outperforms the other algorithms and is the preferred choice for solving the inverse kinematics problem for various manipulators.

Figure 14 illustrates the variation in the objective function for each algorithm over 500 iterations based on results from the first test. As shown in Figure 14, I-BES demonstrates the fastest convergence speed within 500 iterations compared to the other algorithms. During the testing of six different manipulators, I-BES converges to the optimum at around 100 iterations, while BES achieves the same convergence accuracy at around 120 iterations. PSO converges to a suboptimal value but requires 500 iterations, experiencing stagnation several times throughout the process. FA and MTSA show fast convergence speeds when the number of iterations is small; however, as the iterations increase, their convergence speeds gradually slow down, eventually stabilizing at suboptimal values. CPSO-AT, on the other hand, has a slower convergence rate as the number of iterations increases, ultimately converging to a suboptimal value. Consistent with the results in Table 8, it is evident that I-BES and BES outperform the other algorithms.

Figure 15 shows the variation in the best fitness values for each algorithm over 25 tests. As illustrated in Figure 15, as the DOF of the manipulator increases, the best fitness values of both I-BES and BES stabilize below 10^−15^, while the best fitness values of PSO stabilize between 10^−10^ and 10^−5^. The best fitness values for FA, MTSA, and CPSO-AT stabilize between 10^−5^ and 10^0^. According to Figure 15, it is evident that I-BES and BES achieve higher convergence accuracy than PSO, FA, MTSA, and CPSO-AT with the same number of iterations. Additionally, I-BES is more robust than BES and can consistently search for high-accuracy solutions, indicating that I-BES has a greater advantage in solving the inverse kinematics problem.

#### 5.1.3. Comparison against the Geometric Approach

Geometric methods provide exact solutions for inverse kinematics, but these methods often require analyzing complex kinematic structures, especially when dealing with manipulators with high DOF. In this research, the I-BES is compared to the geometric approach, with the primary goal of evaluating the accuracy of the proposed method in relation to exact solutions.

The geometric method compared in this study is from the literature [45]. In the cited work, Kalra et al. studied the Puma 560 manipulator with 6DOF. They established five sets of desired positions for the end-effector, where four viable joint solutions could be obtained. To ensure the objectivity of the comparative tests, the experimental conditions in this study were kept consistent with those in the literature [45]. The specific desired positions are as follows:***p***_*d*_ = [0.60 0.15 0.20]^*T*^.***p***_*d*_ = [0.50 0.24 0.23]^*T*^.***p***_*d*_ = [0.54 0.21 0.26]^*T*^.***p***_*d*_ = [0.18 −0.40 0.40]^*T*^.***p***_*d*_ = [−0.18 0.40 −0.20]^*T*^.

Considering that only the desired position of the end-effector is essential, solving the inverse kinematics for the puma 560 requires consideration of its initial 3DOF. The parameter settings for I-BES remain consistent with those used in Section 5.1.1. The testing procedure was conducted in MATLAB™, with each desired position triggering a single execution of I-BES.

The comparison results are presented in Table 9. According to these results, the joint angles calculated using I-BES show a high degree of overlap with those obtained using the geometric method, with the joint angle error stabilized at 10^−5^, demonstrating the ability of the I-BES to robustly identify multiple joint solutions.

#### 5.1.4. Experiments

The main purpose of the experiments is to demonstrate the effectiveness of the proposed method for solving inverse kinematics under real joint constraints. A comparative analysis involving I-BES, BES, PSO, FA, MTSA, and CPSO-AT is also conducted. The specific parameter settings for the algorithms are consistent with those used in the simulations presented in Section 5.1.1.

The robotic manipulator used in the experiments is the 7DOF rethink sawyer manipulator, as shown in Figure 16. The DH parameters for the manipulator are provided in Table 7. The joint limits of the manipulator are as follows.
(17)$$\begin{array}{c} \left\{\begin{array}{c} q_{l}\left(i\right)=-\pi \;i=1,\;2,\;\ldots ,\;7\\ q_{u}\left(i\right)=\;\pi \;i=1,\;2,\;\ldots ,\;7 \end{array}\right. \end{array}$$

The experiments were conducted using C++ and the robot operating system (ROS). The ROS interacts with the manipulator through an interface called the ’Sawyer SDK.’ The experimental procedures are detailed as follows:

**Experiment 1:** The task involves determining four sets of appropriate joint configurations to reach the specified end-effector position. The desired position is defined as $p_{d}={\left[0.45\;0.45\;0.45\right]}^{T}$, considering the first six DOF of the manipulator.

**Experiment 2:** This task involves determining four sets of appropriate joint configurations to reach the specified end-effector position. The desired position is defined as $p_{d}={\left[0.5\;0.3\;0.1\right]}^{T}$, considering all the DOF of the manipulator.

The reason for using either 6DOF or 7DOF in the experiments is to demonstrate the robustness of the proposed method.

In the experiments, the joint angles derived from each algorithm were applied to the hardware of the rethink sawyer manipulator. The results of Experiment 1 and Experiment 2 are presented in Table 10 and Table 11, respectively.

Table 10 presents the results of Experiment 1. As shown in Table 10, all the compared algorithms can identify the four sets of solutions when the seventh joint angle is fixed at zero. Among these algorithms, I-BES and BES achieve the smallest positional errors, consistently ranging from 0 m to 10^−17^ m, followed by PSO, which maintains a stable positional error of approximately 10^−8^ m.

Table 11 presents the results of Experiment 2. As shown in Table 11, all the compared algorithms successfully find the four sets of solutions. Among these algorithms, I-BES and BES achieve the smallest positional error, which is 0 m. PSO follows, with a stable positional error ranging from 10^−8^ m to 10^−7^ m.

In general, compared to PSO, FA, MTSA, and CPSO-AT, it is evident that I-BES and BES achieve better performance in terms of positional error, maintaining stability within the range of 0 m to 10^−17^ m. Furthermore, the positional accuracy of I-BES and BES remains consistent even as the number of solutions increases, demonstrating their robustness.

For a visual representation, the four joint configurations of I-BES for Experiment 1 and Experiment 2 are shown in Figure 17 and Figure 18, respectively.

### 5.2. CEC2017 Test Results and Analysis

To further validate the performance of I-BES, nine test functions are selected from the CEC2017 test functions. The test results are shown in Table 12. The constraint range of variables during the test is set to [−100, 100]. In the experiment, I-BES is compared with BES, PSO, FA, MTSA, and CPSO-AT. The algorithm parameters are configured as shown in Table 1. The maximum number of iterations is set to *T* = 500, the population size is set to *N* = 30, and the dimension is set to *D* = 30. Each algorithm is run independently 30 times. The performance of the algorithms is evaluated based on the optimum value (best), average value (mean), standard deviation (Std), and convergence curves. Additionally, the Wilcoxon rank-sum test is used to verify the significance of differences between I-BES and the comparison algorithms.

Table 12 shows the statistical results of the 6 algorithms after 30 runs, with the optimal results highlighted in bold. Here, F3 is a single-peak function used to evaluate the convergence speed and local search ability of the algorithms. As can be seen from Table 12, for the single-peak function F3, I-BES outperforms the other five algorithms and demonstrates a noticeable improvement compared to BES, indicating that I-BES possesses excellent local search capabilities.

Multi-peak test functions are used to evaluate the performance of the algorithm in solving multi-peak complex optimization problems. Among them, functions F8 and F9 are specifically designed to evaluate the ability of the algorithm to perform global searches and escape from local optimal solutions. According to Table 12, I-BES achieves the best average values on F8 and F9 compared to the other algorithms. This result demonstrates that I-BES has a strong global search capability and effectively escapes from local optima.

A hybrid function consists of several different functions that are combined to create variety and complexity. A composite function is formed by combining multiple simple functions. Each simple function is a univariate function, but when combined, they form a high-dimensional composite function. These two types of functions can be used to evaluate the tolerance of an algorithm to noise and infeasible solutions, as well as its ability to solve large-scale optimization problems. I-BES demonstrates significant advantages in the six hybrid and composite functions, F14, F15, F18, F19, F27, and F30, achieving the lowest mean and optimal values among all the compared algorithms. In addition, the standard deviation of I-BES is second only to that of MTSA and CPSO-AT in F19 and F27, and it ranks first among the remaining four tested functions. This demonstrates the effectiveness of I-BES in solving high-dimensional problems.

To further evaluate the performance of I-BES, the Wilcoxon rank-sum test was used for validation [46,47]. The Wilcoxon signed-rank test was performed to compare the results of I-BES with those of the other five algorithms at a significance level of α = 5%. Generally, a *p*-value of less than 0.05 indicates a significant difference between the two sets of data, while “N/A” indicates that the algorithms produce similar results. Column R in the Wilcoxon signed-rank test table shows the test results, where the symbols “+”, “−”, and “=” indicate that the performance of I-BES is better than, worse than, or equal to the comparison algorithm, respectively.

Table 13 presents the convergence curves of the algorithms on the CEC2017 test functions. According to Table 13, there is no significant difference between I-BES and PSO for functions F8 and F9, while a significant difference exists between I-BES and the other four algorithms. The R column indicates that for most of the functions, I-BES outperforms the comparison algorithms. The results of the Wilcoxon rank-sum test demonstrate that, statistically, I-BES performs better than the original algorithm and the four comparison algorithms on the CEC2017 test functions.

Figure 19 presents the convergence curves of the six algorithms when testing the CEC2017 function over 500 iterations. As shown in Figure 19, I-BES exhibits a significant advantage in both convergence speed and accuracy. I-BES achieves a lower fitness value compared to BES, which demonstrates the effectiveness of the Lévy flight strategy. For F3, after the 50th iteration, I-BES converges faster than BES, as the Lévy flight strategy accelerates the convergence of the algorithm. In addition, for F30, it can be observed that the fitness value of BES remains almost unchanged after several iterations, while I-BES continues to search for a better solution. This demonstrates the capability of the Lévy flight strategy to escape from local optima.

The experimental results of the CEC2017 test functions demonstrate that I-BES performs well and is highly effective in solving complex problems. For hybrid and composite functions, I-BES exhibits optimal performance. The improved algorithm shows good convergence speed and accuracy, as well as the ability to avoid falling into local optima, which is of great significance for solving practical optimization problems.

### 5.3. Other Engineering Problems

In this section, two real-world engineering optimization problems are used to evaluate the practicality and reliability of I-BES. The population size of the algorithm is set to *N* = 30 for each engineering problem. The maximum number of iterations is set to *T* = 500, and the algorithm is run independently 30 times for comparison.

#### 5.3.1. Tension/Compression Spring Design Problem

This tension/compression spring design problem consists of three variables: $x_{1}$ (wire diameter, *d*), $x_{2}$ (mean coil diameter, *D*), and $x_{3}$ (number of active coils, *P*). The objective of this problem is to minimize the weight (*f*) of a tension/compression spring, as shown in Figure 20. The mathematical formulation of the tension/compression spring design problem is as follows:
(18)$$f\left(x\right)=\left(x_{3}+2\right)x_{2}x_{1}^{2}$$
$$x=\left[x_{1},x_{2},x_{3}\right]=\left[d,D,P\right]$$
$$\left\{\begin{array}{c} g_{1}\left(x\right)=1-\frac{x_{2}^{3}x_{3}}{71.785x_{1}^{4}}\leq 0\\ g_{2}\left(x\right)=\frac{4x_{2}^{2}-x_{1}x_{2}}{12.566(x_{2}x_{1}^{3}-x_{1}^{4})}+\frac{1}{5.108x_{1}^{2}}-1\leq 0\\ g_{3}\left(x\right)=1-\frac{140.45x_{1}}{x_{2}^{2}x_{3}}\leq 0\\ g_{4}\left(x\right)=\frac{x_{1}+x_{2}}{1.5}-1\leq 0 \end{array}\right.$$
$$0.05\leq x_{1}\leq 2,\;0.25\leq x_{2}\leq 1.3,\;2\leq x_{3}\leq 15,$$

Table 14 shows the best results of the tension/compression spring design problem after 30 runs. The results indicate that I-BES performs better compared to BES, as I-BES, PSO, and MTSA all provide the optimal solution with a function value of $2.6952\times 10^{6}$. In contrast, BES provides a sub-optimal solution and ranks last, with a function value of $2.8151\times 10^{6}$. Table 15 presents the statistical results for all the algorithms, which further support the above findings.

#### 5.3.2. Pressure Vessel Design Problem

This pressure vessel design problem consists of four variables: $x_{1}$ (shell thickness, $T_{s}$), $x_{2}$ (head thickness, $T_{h}$), $x_{3}$ (inner radius, *R*), and $x_{4}$ (length of the cylindrical section, *L*). The objective of this problem is to minimize the total cost (*f*) of the pressure vessel, including material, forming, and welding costs, as shown in Figure 21. The mathematical formulation of the pressure vessel design problem is as follows:
(19)$$f\left(x\right)=0.6224x_{1}x_{3}x_{4}+1.7781x_{2}x_{3}^{2}+3.1661x_{1}^{2}x_{4}+19.84x_{1}^{2}x_{3}$$
$$x=\left[x_{1},x_{2},x_{3},x_{4}\right]=\left[T_{s},T_{h},R,L\right]$$
$$\left\{\begin{array}{c} g_{1}\left(x\right)=-x_{1}+0.0193x_{3}\leq 0\\ g_{2}\left(x\right)=-x_{2}+0.00954x_{3}\leq 0\\ g_{3}\left(x\right)=-\pi x_{3}^{2}x_{4}-\frac{4}{3}\pi x_{3}^{3}+1296000\leq 0\\ g_{4}\left(x\right)=x_{4}-240\leq 0 \end{array}\right.$$
$$0\leq x_{1,2}\leq 99,\;10\leq x_{3,4}\leq 200$$

Table 16 shows the optimal results for the pressure vessel design problem. According to Table 16, I-BES provides the optimal solution for this problem, achieving a minimum cost of $5.8862\times 10^{3}$. The total cost of the pressure vessel is minimized when $T_{s}$, $T_{h}$, *R*, and *L* are set to 0.7782, 0.3847, 40.3223, and 200, respectively. The MTSA algorithm ranks second in this problem, while BES ranks fifth. Table 17 presents the statistical results of all the algorithms. It is also evident from Table 17 that I-BES demonstrates a significant performance improvement over BES.

### 5.4. Discussion

This paper presents an inverse kinematics solution method based on the improved bald eagle search optimization algorithm. Through simulations and experimental results of 2DOF to 7DOF manipulators, I-BES demonstrates a faster convergence speed, higher accuracy, and greater robustness compared to other metaheuristic algorithms such as PSO, FA, MTSA, and CPSO-AT. These advantages are particularly evident in solving nonlinear problems and improving search efficiency.

The simulation and experimental results for the inverse kinematics of manipulators indicate that I-BES outperforms tje other methods in controlling the position error of the end-effector, as shown in Figure 8, Figure 9, Figure 10, Figure 11, Figure 12 and Figure 13 and Table 8. Furthermore, I-BES achieves faster convergence with the same number of iterations (Figure 14), and its robustness is confirmed through multiple tests on different manipulators, demonstrating a reliable ability to locate the global optimum (Figure 15). Compared to the geometric method, I-BES also shows the ability to identify multiple joint configurations with high positional accuracy (Table 9). In real-world experiments, I-BES consistently outperforms other algorithms, as demonstrated in Table 10 and Table 11. Figure 17 and Figure 18 further illustrate the feasibility of I-BES under real joint constraints.

The results of the CEC2017 test set show that I-BES significantly outperforms other algorithms across various functions, as detailed in Table 12. Specifically, I-BES demonstrates superior local search capability on the single-peak function F3. For the multi-peak functions F8 and F9, I-BES achieves the best average values, reflecting its strong global search and ability to escape local optima. In the hybrid and composite functions (F14, F15, F18, F19, F27, F30), I-BES shows significant advantages by achieving the lowest mean and optimal values. Moreover, I-BES attains the top performance in terms of standard deviation for most functions, further validating its robustness in high-dimensional optimization problems. The Wilcoxon rank-sum test results in Table 13 indicate that I-BES significantly outperforms all comparison algorithms, except for PSO on functions F8 and F9.

The results of the other two engineering design problems show that I-BES achieves superior performance in both the tension/compression spring design and pressure vessel design problems. As shown in Table 14, I-BES provides the optimal solution for the spring design problem, demonstrating improved performance over BES. For the pressure vessel design problem (Table 16), I-BES achieves the lowest cost, outperforming both MTSA and BES.

The I-BES incorporating the Lévy flight strategy offers significant advantages over the original BES in terms of global search capability, convergence accuracy, and the ability to escape local optima. The Lévy flight strategy utilizes its long-distance jump characteristics to prevent the method from prematurely converging to local optima during complex function optimization, thereby enhancing global search efficiency. When solving the inverse kinematics of a manipulator, I-BES can more effectively locate the global optimum, reduce solution accuracy errors, and exhibit better robustness when dealing with complex joint configurations and high degrees of redundancy. Furthermore, in engineering applications, I-BES demonstrates superior performance in constrained optimization and multi-objective design problems. For example, in practical engineering optimization tasks such as spring design and pressure vessel design, I-BES achieves optimal solutions with lower computational cost, verifying its potential and reliability in real-world engineering applications.

Future work will focus on applying I-BES to more complex robotic systems, such as mobile manipulators and dual-arm manipulator systems, to broaden its practical applications. By integrating real-time feedback and adaptive strategies, I-BES can be further refined to handle dynamic and uncertain operational environments. These advancements will enable I-BES to perform more effectively in scenarios requiring high flexibility and precision, thereby facilitating its deployment in diverse robotic applications [48,49].

## 6. Conclusions

In this paper, an improved bald eagle search optimization method is proposed to enhance global search efficiency by introducing the Lévy flight strategy in the search phase of the original method, thereby preventing it from falling into local optima prematurely. Experimental results show that the convergence accuracy of I-BES can reach $10^{-18}$ m when solving the inverse kinematics problem of manipulators. Moreover, I-BES outperforms BES and four other popular optimization methods on CEC2017 test functions in terms of search accuracy, convergence speed, and stability. Compared to BES, I-BES improves performance on most test functions by more than 70%. Finally, two additional case studies involving engineering optimization problems further validate that I-BES achieves better optimization results in real-world scenarios. Future research will focus on exploring its application to mobile and dual-arm manipulators. 

## Figures and Tables

**Figure 1 biomimetics-09-00627-f001:**
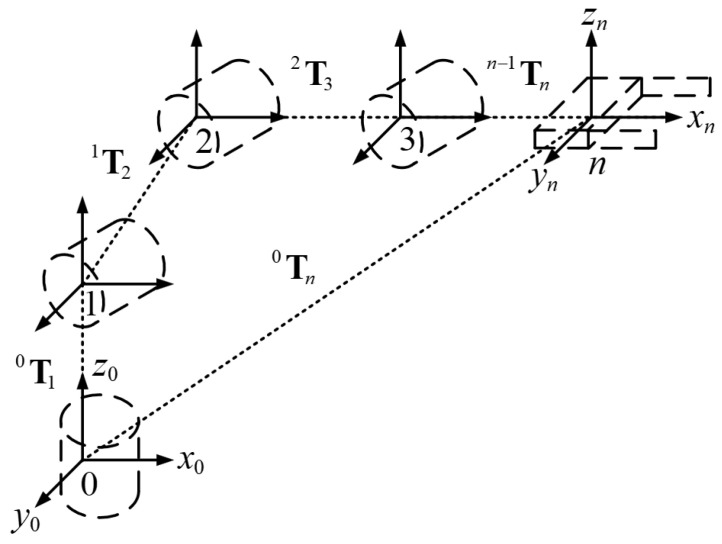
The kinematic chain of a serial robotic manipulator.

**Figure 2 biomimetics-09-00627-f002:**
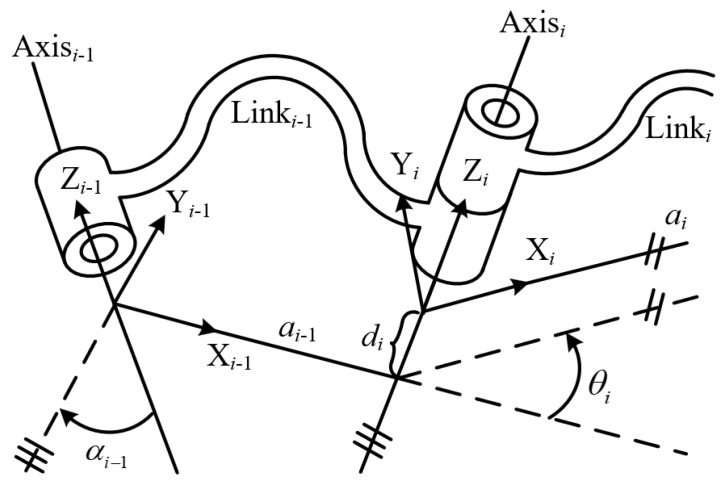
Correspondence of link parameters between the (*i* − 1)-th link and the *i*-th link of the manipulator.

**Figure 3 biomimetics-09-00627-f003:**
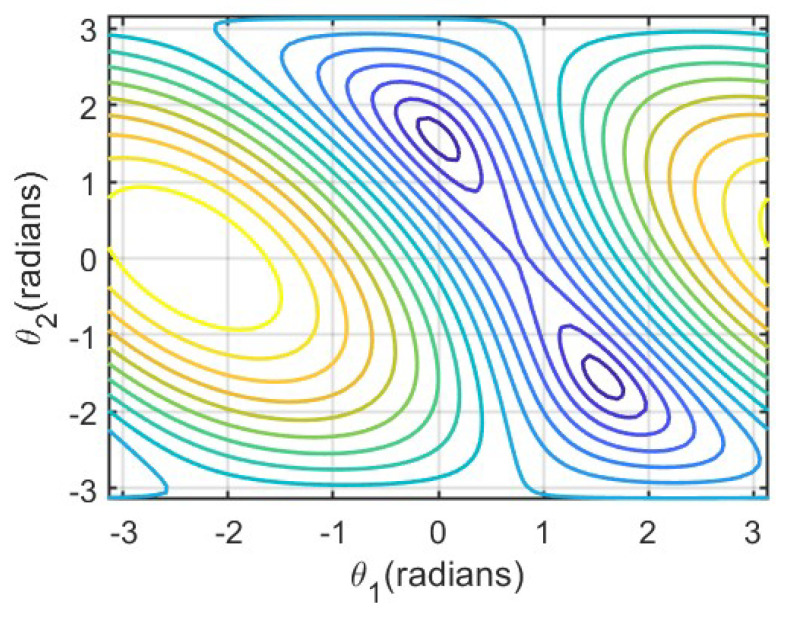
The error function of the IK problem for a 2DOF planar manipulator.

**Figure 4 biomimetics-09-00627-f004:**
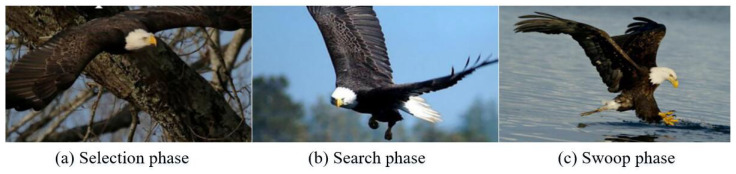
Bald eagle prey hunting process.

**Figure 5 biomimetics-09-00627-f005:**
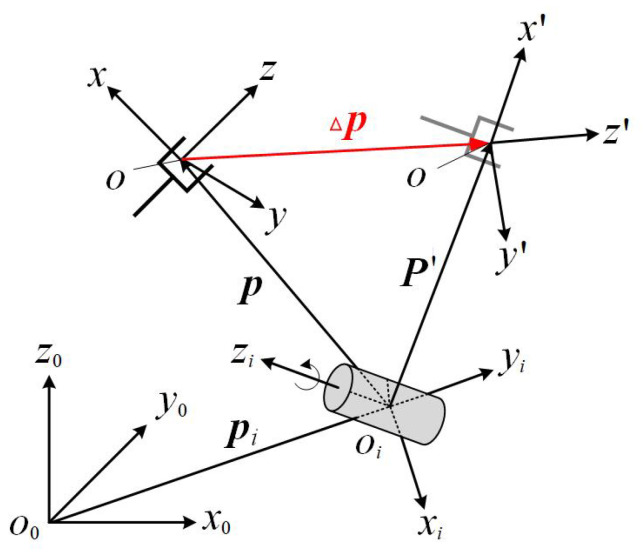
Position error schematic.

**Figure 6 biomimetics-09-00627-f006:**
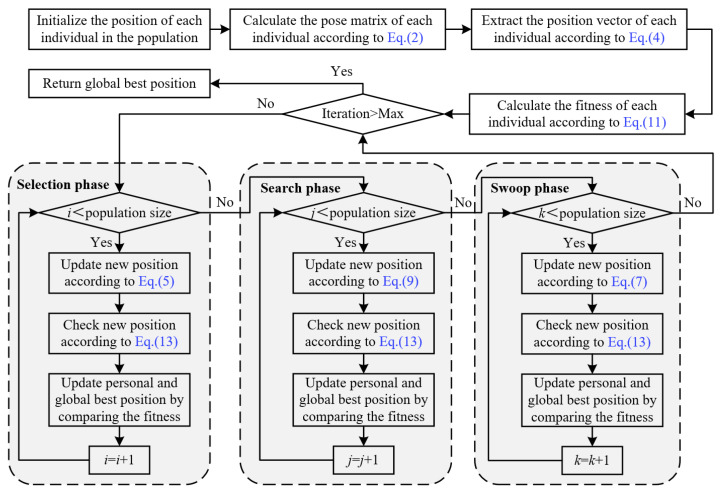
Flowchart for solving the inverse kinematics of robotic manipulators using I-BES.

**Figure 7 biomimetics-09-00627-f007:**
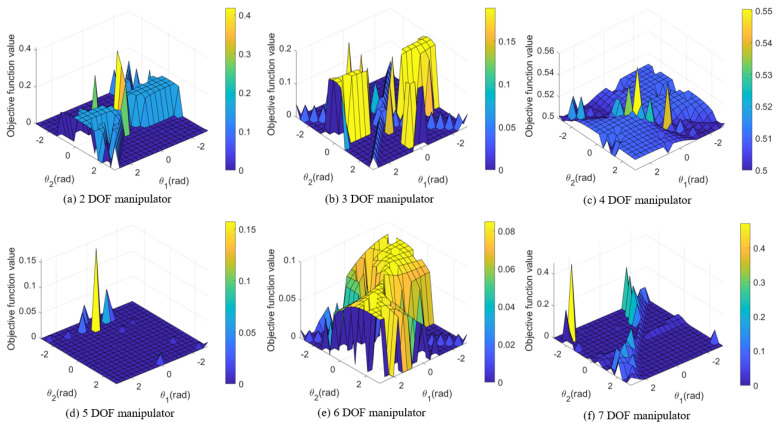
Variations in the objective function value about the first two joint angles in manipulators of 2DOF to 7DOF.

**Figure 8 biomimetics-09-00627-f008:**
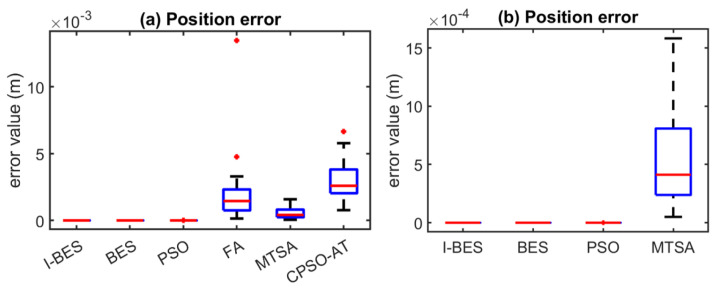
Position errors of 2DOF manipulator.

**Figure 9 biomimetics-09-00627-f009:**
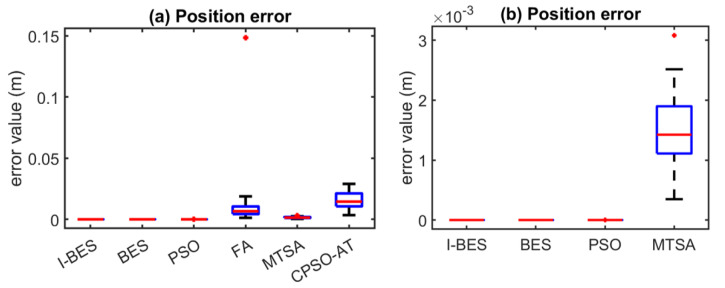
Position errors of 3DOF manipulator.

**Figure 10 biomimetics-09-00627-f010:**
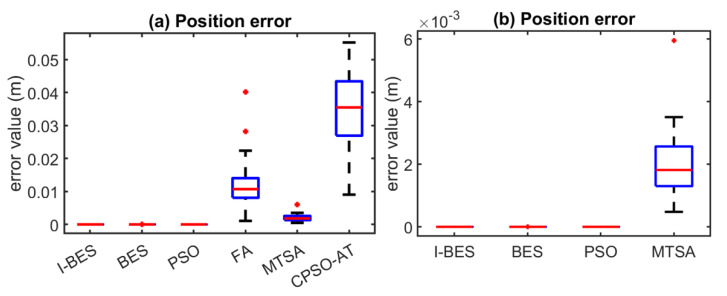
Position errors of 4DOF manipulator.

**Figure 11 biomimetics-09-00627-f011:**
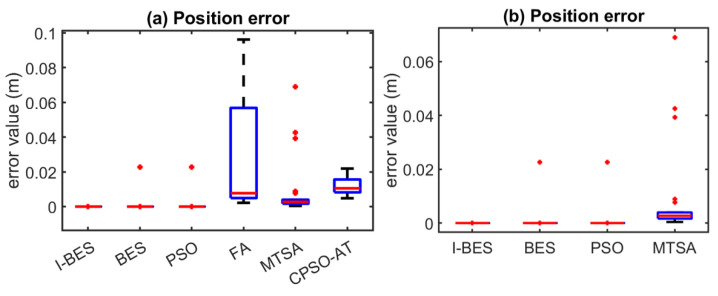
Position errors of 5DOF manipulator.

**Figure 12 biomimetics-09-00627-f012:**
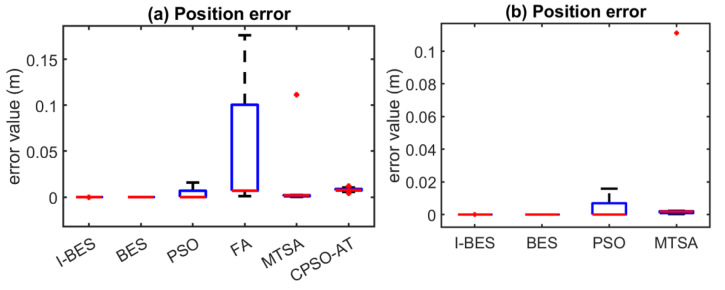
Position errors of 6DOF manipulator.

**Figure 13 biomimetics-09-00627-f013:**
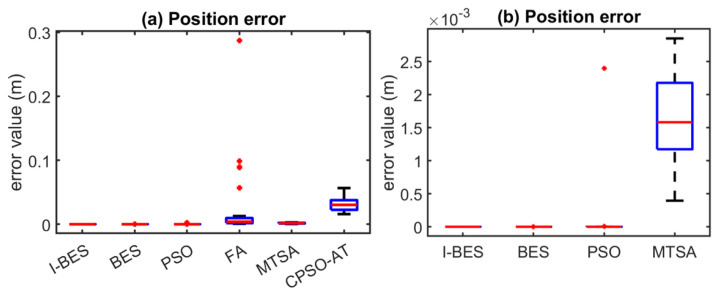
Position errors of 7DOF manipulator.

**Figure 14 biomimetics-09-00627-f014:**
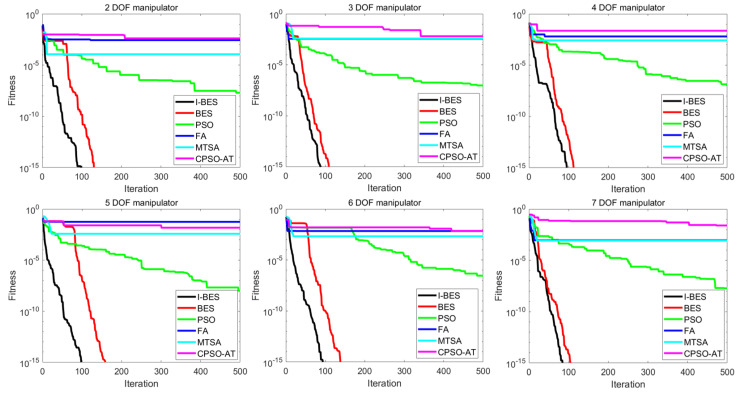
Variations in the objective function for each algorithm over 500 iterations.

**Figure 15 biomimetics-09-00627-f015:**
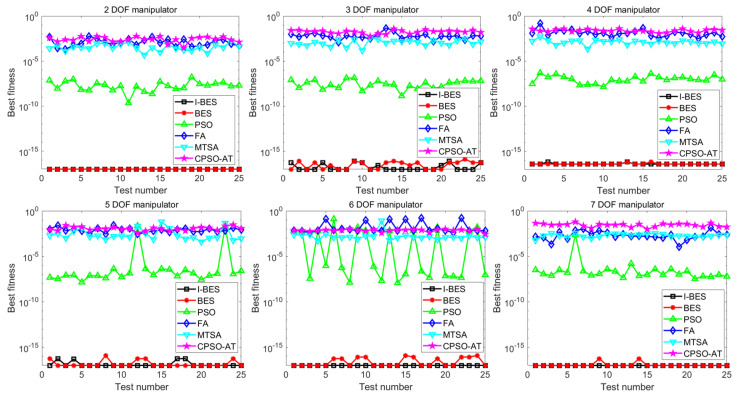
Variations in the best fitness for each algorithm over 25 tests.

**Figure 16 biomimetics-09-00627-f016:**
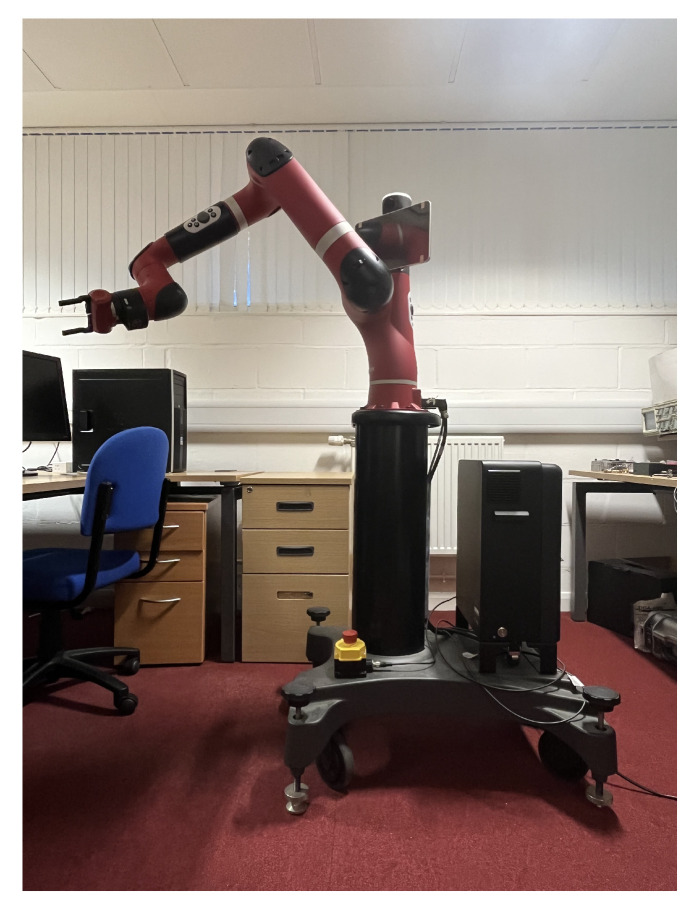
Schematic diagram of 7DOF manipulator.

**Figure 17 biomimetics-09-00627-f017:**
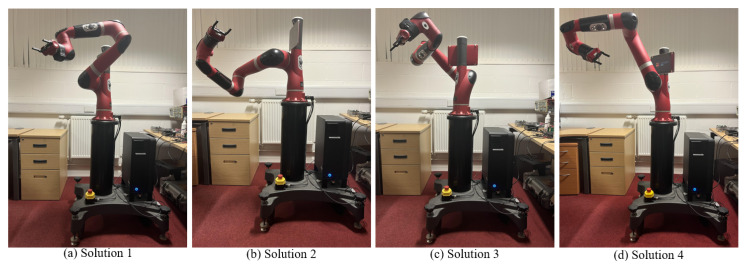
Four joint configurations of I-BES for Experiment 1.

**Figure 18 biomimetics-09-00627-f018:**
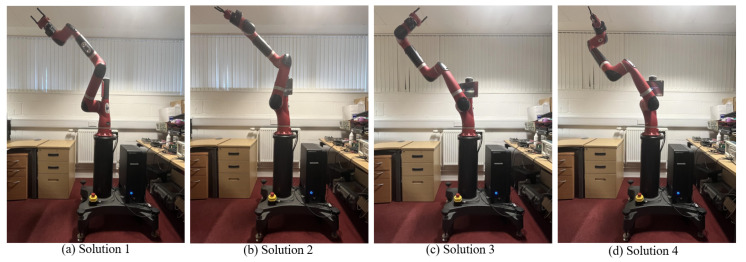
Four joint configurations of I-BES for Experiment 2.

**Figure 19 biomimetics-09-00627-f019:**
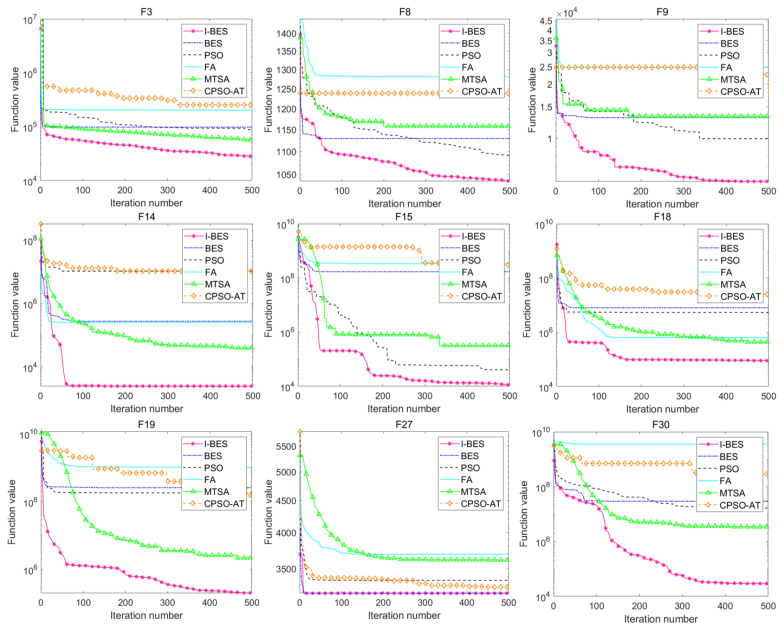
Convergence curves of algorithms on CEC2017 test functions.

**Figure 20 biomimetics-09-00627-f020:**
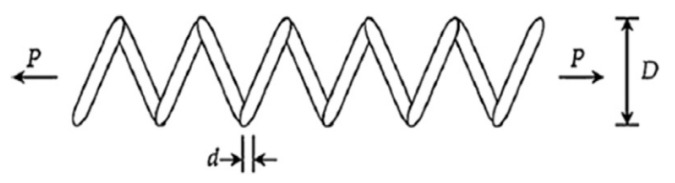
Tension/compression spring design.

**Figure 21 biomimetics-09-00627-f021:**
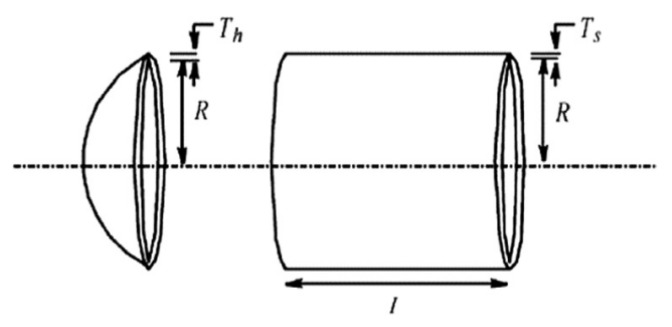
Pressure vessel design.

**Table 1 biomimetics-09-00627-t001:** Parameter configurations of each algorithm.

Algorithm	Parameter
I-BES	$c_{1}=1.5,c_{2}=1.5,R=1.4,\alpha =1.8,a=8$
BES	$c_{1}=1.5,c_{2}=1.5,R=1.4,\alpha =1.8,a=8$
PSO	$c_{1}=1.4,c_{2}=1.4,v=0.5x,\omega =0.9$
FA	$\alpha =0.2,\beta_{0}=2,\gamma =1$
MTSA [43]	$S_{max}=3,\epsilon =0.01$
CPSO-AT [44]	$\omega_{max}=0.9,\omega_{min}=0.4,c_{1}=2,c_{2}=2$

**Table 2 biomimetics-09-00627-t002:** DH parameters of the 2DOF manipulator.

Joint	θ (rad)	*d* (m)	*a* (m)	α (rad)
1	θ _1_	0.0	0.35	0.0
2	θ _2_	0.0	0.30	0.0

**Table 3 biomimetics-09-00627-t003:** DH parameters of the 3DOF manipulator.

Joint	θ (rad)	*d* (m)	*a* (m)	α (rad)
1	θ _1_	0.35	0.00	π/2
2	θ _2_	0.00	0.30	0.0
3	θ _3_	0.00	0.25	0.0

**Table 4 biomimetics-09-00627-t004:** DH parameters of the 4DOF manipulator.

Joint	θ (rad)	*d* (m)	*a* (m)	α (rad)
1	θ _1_	0.8	0.15	0.0
2	θ _2_	0.0	0.40	π
3	0	*d* _3_	0.00	0.0
4	θ _4_	0.2	0.00	0.0

**Table 5 biomimetics-09-00627-t005:** DH parameters of the 5DOF manipulator.

Joint	θ (rad)	*d* (m)	*a* (m)	α (rad)
1	θ _1_	0.147	0.033	π/2
2	θ _2_	0.000	0.155	0.0
3	θ _3_	0.000	0.135	0.0
4	θ _4_	0.000	0.000	π/2
5	θ _5_	0.218	0.000	0.0

**Table 6 biomimetics-09-00627-t006:** DH parameters of the 6DOF manipulator.

Joint	θ (rad)	*d* (m)	*a* (m)	α (rad)
1	θ _1_	0.000	0.000	π/2
2	θ _2_	0.000	0.431	0.0
3	θ _3_	0.150	0.020	−π/2
4	θ _4_	0.432	0.000	π/2
5	θ _5_	0.000	0.000	−π/2
6	θ _6_	0.000	0.000	0.0

**Table 7 biomimetics-09-00627-t007:** DH parameters of the 7DOF manipulator.

Joint	θ (rad)	*d* (m)	*a* (m)	α (rad)
1	θ _1_	0.3170	0.081	−π/2
2	θ _2_	0.1925	0.000	π/2
3	θ _3_	0.4000	0.000	π/2
4	θ _4_	0.1685	0.000	−π/2
5	θ _5_	0.4000	0.000	−π/2
6	θ _6_	0.1363	0.000	π/2
7	θ _7_	0.1338	0.000	0.0

**Table 8 biomimetics-09-00627-t008:** Position errors (m) of algorithms.

Simulation	Position Error	I-BES	BES	PSO	FA	MTSA	CPSO-AT
2DOF manipulator	Average	**0**	0	1.84 × 10^−8^	2.06 × 10^−3^	5.66 × 10^−4^	2.93 × 10^−3^
	Median	**0**	0	1.24 × 10^−8^	1.45 × 10^−3^	4.12 × 10^−4^	2.59 × 10^−3^
	Min	**0**	0	5.56 × 10^−10^	1.39 × 10^−4^	4.99 × 10^−5^	7.68 × 10^−4^
	Max	**0**	0	8.43 × 10^−8^	1.35 × 10^−2^	1.58 × 10^−3^	6.66 × 10^−3^
3DOF manipulator	Average	**3.77 × 10^−17^**	4.08 × 10^−17^	6.61 × 10^−8^	1.30 × 10^−2^	1.53 × 10^−3^	1.53 × 10^−2^
	Median	**5.55 × 10^−17^**	5.55 × 10^−17^	5.85 × 10^−8^	6.63 × 10^−3^	1.42 × 10^−3^	1.45 × 10^−2^
	Min	**0**	0	2.93 × 10^−9^	1.24 × 10^−3^	3.48 × 10^−4^	3.44 × 10^−3^
	Max	**7.85 × 10^−17^**	1.24 × 10^−16^	2.21 × 10^−7^	1.49 × 10^−1^	3.08 × 10^−3^	2.90 × 10^−2^
4DOF manipulator	Average	**3.93 × 10^−17^**	4.22 × 10^−17^	1.50 × 10^−7^	1.22 × 10^−2^	2.05 × 10^−3^	3.42 × 10^−2^
	Median	**3.93 × 10^−17^**	3.93 × 10^−17^	1.25 × 10^−7^	1.07 × 10^−2^	1.82 × 10^−3^	3.55 × 10^−2^
	Min	**3.93 × 10^−17^**	3.93 × 10^−17^	4.63 × 10^−9^	1.07 × 10^−3^	4.77 × 10^−4^	9.03 × 10^−3^
	Max	**3.93 × 10^−17^**	6.80 × 10^−17^	4.10 × 10^−7^	4.01 × 10^−2^	5.95 × 10^−3^	5.52 × 10^−2^
5DOF manipulator	Average	**1.31 × 10^−17^**	9.05 × 10^−4^	9.05 × 10^−4^	3.11 × 10^−2^	8.49 × 10^−3^	1.20 × 10^−2^
	Median	**0**	0	6.55 × 10^−8^	7.70 × 10^−3^	2.72 × 10^−3^	1.05 × 10^−2^
	Min	**0**	0	4.40 × 10^−9^	2.16 × 10^−3^	3.81 × 10^−4^	4.80 × 10^−3^
	Max	**5.55 × 10^−17^**	2.26 × 10^−2^	2.26 × 10^−2^	9.61 × 10^−2^	6.90 × 10^−2^	2.19 × 10^−2^
6DOF manipulator	Average	**2.22 × 10^−18^**	3.55 × 10^−17^	4.56 × 10^−3^	4.24 × 10^−2^	5.92 × 10^−3^	8.12 × 10^−3^
	Median	**0**	0	2.21 × 10^−7^	7.00 × 10^−3^	1.84 × 10^−3^	7.76 × 10^−3^
	Min	**0**	0	6.11 × 10^−9^	1.07 × 10^−3^	2.89 × 10^−4^	4.03 × 10^−3^
	Max	**5.55 × 10^−17^**	8.33 × 10^−17^	1.58 × 10^−2^	1.76 × 10^−1^	1.11 × 10^−1^	1.20 × 10^−2^
7DOF manipulator	Average	**0**	4.44 × 10^−18^	9.64 × 10^−5^	2.80 × 10^−2^	1.67 × 10^−3^	3.17 × 10^−2^
	Median	**0**	0	1.62 × 10^−7^	3.86 × 10^−3^	1.58 × 10^−3^	3.02 × 10^−2^
	Min	**0**	0	1.28 × 10^−8^	8.02 × 10^−4^	3.95 × 10^−4^	1.58 × 10^−2^
	Max	**0**	1.11 × 10^−16^	2.40 × 10^−3^	2.87 × 10^−1^	2.85 × 10^−3^	5.66 × 10^−2^

**Table 9 biomimetics-09-00627-t009:** Comparison results between the I-BES and the geometric approach.

Test	Method	θ_1_ (rad)	θ_2_ (rad)	θ_3_ (rad)	$∥{q_{i}}^{g}-{q_{i}}^{I-\mathrm{BES}}∥$_2_ ^1^
1	$\left({q_{1}}^{g}\right)$ ^2^	0	−1.0748	3.1201	**3.98 × 10^−5^**
	$\left({q_{1}}^{\text{I-BES}}\right)$ ^3^	0	−1.07483809	3.12008845	
	q _2_ ^ *g* ^	0	0.4313	0.1153	**4.35 × 10^−5^**
	q _2_ ^ *I-BES* ^	0	0.43133699	0.11527708	
	q _3_ ^ *g* ^	−2.6545	−2.0668	0.1153	**5.16 × 10^−5^**
	q _3_ ^ *I-BES* ^	−2.65449125	−2.06675456	0.11527708	
	q _4_ ^ *g* ^	−2.6545	2.7103	3.1201	**4.66 × 10^−5^**
	q _4_ ^ *I-BES* ^	−2.65449125	2.71025567	3.12008845	
2	${q_{1}}^{g}$	0.1754	−1.2425	−2.998	**5.58 × 10^−5^**
	${q_{1}}^{\text{I-BES}}$	0.17535625	−1.24252231	−2.99802640	
	${q_{2}}^{g}$	0.1754	0.4294	−0.0498	**4.43 × 10^−5^**
	${q_{2}}^{\text{I-BES}}$	0.17535625	0.42940065	−0.04979338	
	${q_{3}}^{g}$	−2.4219	−1.8991	−0.0498	**3.17 × 10^−5^**
	${q_{3}}^{\text{I-BES}}$	−2.42190895	−1.89907034	−0.04979338	
	${q_{4}}^{g}$	−2.4219	2.7122	−2.998	**2.90 × 10^−5^**
	${q_{4}}^{\text{I-BES}}$	−2.42190895	2.71219200	−2.99802640	
3	${q_{1}}^{g}$	0.1106	−1.2132	−3.1125	**5.78 × 10^−5^**
	${q_{1}}^{\text{I-BES}}$	0.11064393	−1.21323656	−3.11249085	
	${q_{2}}^{g}$	0.1106	0.3437	0.0647	**6.63 × 10^−5^**
	${q_{2}}^{\text{I-BES}}$	0.11064393	0.34374040	0.06467106	
	${q_{3}}^{g}$	−2.5105	−1.9284	0.0647	**6.99 × 10^−5^**
	${q_{3}}^{\text{I-BES}}$	−2.51045401	−1.92835609	0.06467106	
	${q_{4}}^{g}$	−2.5105	2.7979	−3.1125	**6.69 × 10^−5^**
	${q_{4}}^{\text{I-BES}}$	−2.51045401	2.79785225	−3.11249085	
4	${q_{1}}^{g}$	−1.4947	−1.6169	−2.9763	**6.19 × 10^−5^**
	${q_{1}}^{\text{I-BES}}$	−1.49474865	−1.61688215	−2.97626623	
	${q_{2}}^{g}$	−1.4947	0.0769	−0.0716	**6.73 × 10^−5^**
	${q_{2}}^{\text{I-BES}}$	−1.49474865	0.07689939	−0.07155356	
	${q_{3}}^{g}$	2.3405	−1.5247	−0.0716	**6.44 × 10^−5^**
	${q_{3}}^{\text{I-BES}}$	2.34045650	−1.52471050	−0.07155356	
	${q_{4}}^{g}$	2.3405	3.0647	−2.9763	**5.54 × 10^−5^**
	${q_{4}}^{\text{I-BES}}$	2.34045650	3.06469327	−2.97626623	
5	${q_{1}}^{g}$	1.6468	−0.5643	−2.6406	**6.58 × 10^−5^**
	${q_{1}}^{\text{I-BES}}$	1.64684400	−0.56425178	−2.64059151	
	${q_{2}}^{g}$	1.6468	1.4671	−0.4072	**5.64 × 10^−5^**
	${q_{2}}^{\text{I-BES}}$	1.64684400	1.46712123	−0.40722827	
	${q_{3}}^{g}$	−0.8011	−2.5773	−0.4072	**6.14 × 10^−5^**
	${q_{3}}^{\text{I-BES}}$	−0.80113615	−2.57734087	−0.40722827	
	${q_{4}}^{g}$	−0.8011	1.6745	−2.6406	**4.69 × 10^−5^**
	${q_{4}}^{\text{I-BES}}$	−0.80113615	1.67447142	−2.64059151	

^1^***q**_i_^g^* represents the solution of the geometric approach for the *i*-th desired position. ^2^
***q**_i_^I-BES^* represents the solution of I-BES for the *i*-th desired position. ^3^ ‖ · ‖_2_ represents the Euclidean norm.

**Table 10 biomimetics-09-00627-t010:** Solution results of inverse kinematics for Experiment 1.

Solution	Method	θ_1_ (rad)	θ_2_ (rad)	θ_3_ (rad)	θ_4_ (rad)	θ_5_ (rad)	θ_6_ (rad)	θ_7_ (rad)	ϵ (m)
1	$\left({q_{1}}^{\text{I-BES}}\right)$ ^1^	**−0.7840**	**−1.2296**	**1.4555**	**−2.2666**	**0.2466**	**−0.3275**	**0**	**0**
	$\left({q_{1}}^{\mathrm{BES}}\right)$ ^1^	−1.9872	−0.6172	−0.1962	1.6635	−0.1810	1.1049	0	0
	$\left({q_{1}}^{\mathrm{PSO}}\right)$ ^1^	−1.6052	−1.9082	3.0000	1.0313	2.6854	3.0000	0	5.38 × 10^−8^
	$\left({q_{1}}^{\mathrm{FA}}\right)$ ^1^	−1.5434	−1.9270	3.0000	1.4548	3.0000	−2.9294	0	3.40 × 10^−3^
	$\left({q_{1}}^{\mathrm{MTSA}}\right)$ ^1^	−0.4070	0.3442	1.8353	−1.6701	−2.5153	−1.7308	0	1.02 × 10^−3^
	$\left({q_{1}}^{\text{CPSO-AT}}\right)$ ^1^	−0.2541	2.7537	3.0000	−2.5152	2.9141	−0.1387	0	1.17 × 10^−2^
2	${q_{2}}^{\text{I-BES}}$	**−0.3438**	**1.7126**	**2.0321**	**−1.8308**	**0.0327**	**1.1171**	**0**	**5.55 × 10** ^ **−17** ^
	${q_{2}}^{\mathrm{BES}}$	−0.4354	2.3286	−1.9772	2.5229	−2.4320	0.8198	0	5.55 × 10^−17^
	${q_{2}}^{\mathrm{PSO}}$	1.1353	0.9509	−0.9867	−1.7079	0.2350	−1.3380	0	9.67 × 10^−8^
	${q_{2}}^{\mathrm{FA}}$	0.0645	2.0788	3.0000	−1.4038	−0.9976	2.2679	0	2.40 × 10^−3^
	${q_{2}}^{\mathrm{MTSA}}$	1.0390	0.8260	2.0853	1.7352	−0.1607	1.4346	0	2.61 × 10^−3^
	${q_{2}}^{\text{CPSO-AT}}$	0.1808	0.7868	−2.6708	1.9031	−0.4034	3.0000	0	2.81 × 10^−2^
3	${q_{3}}^{\text{I-BES}}$	**−2.3998**	**−0.8270**	**−1.9257**	**−1.3778**	**1.4765**	**−1.7471**	**0**	**0**
	${q_{3}}^{\mathrm{BES}}$	−0.8852	−2.2387	1.8684	−2.2913	2.9485	−0.5001	0	5.55 × 10^−17^
	${q_{3}}^{\mathrm{PSO}}$	−0.9691	0.4521	−1.7370	2.2981	3.0000	−0.0272	0	4.08 × 10^−8^
	${q_{3}}^{\mathrm{FA}}$	−0.1591	2.0736	3.0000	−1.7855	−1.5423	2.0461	0	1.10 × 10^−3^
	${q_{3}}^{\mathrm{MTSA}}$	−2.3928	−0.5910	−2.0820	−1.3952	−2.5836	1.1428	0	5.48 × 10^−4^
	${q_{3}}^{\text{CPSO-AT}}$	−0.2499	1.1136	2.2887	−2.3338	2.3221	2.7409	0	3.44 × 10^−2^
4	${q_{4}}^{\text{I-BES}}$	**−0.1478**	**0.7685**	**−2.3171**	**1.3883**	**0.7693**	**−2.4884**	**0**	**0**
	${q_{4}}^{\mathrm{BES}}$	0.7584	1.6894	0.6379	0.8997	0.8899	2.5305	0	0
	${q_{4}}^{\mathrm{PSO}}$	0.2556	2.2143	−0.9714	1.4364	1.4391	−2.3286	0	8.54 × 10^−8^
	${q_{4}}^{\mathrm{FA}}$	−1.2686	−0.8141	2.8267	−1.8202	3.0000	3.0000	0	5.80 × 10^−3^
	${q_{4}}^{\mathrm{MTSA}}$	−1.1945	−2.1724	−2.8089	1.6703	1.3771	−1.3786	0	2.93 × 10^−3^
	${q_{4}}^{\text{CPSO-AT}}$	−2.1405	−2.2231	0.5641	−0.9824	−1.6159	1.4473	0	5.17 × 10^−2^

^1^***q**_i_^I-BES^*, ^1^***q**_i_^BES^*, ^1^***q**_i_^PSO^*, ^1^***q**_i_^FA^*, ^1^***q**_i_^MTSA^*, and ^1^***q**_i_^CPSO-AT^* represent the *i*-th set of solutions obtained based on I-BES, BES, PSO, FA, MTSA and CPSO-AT.

**Table 11 biomimetics-09-00627-t011:** Solution results of inverse kinematics for Experiment 2.

Solution	Method	θ_1_ (rad)	θ_2_ (rad)	θ_3_ (rad)	θ_4_ (rad)	θ_5_ (rad)	θ_6_ (rad)	θ_7_ (rad)	ϵ (m)
1	${q_{1}}^{\text{I-BES}}$	**−1.0100**	**−0.4371**	**1.1140**	**−0.9186**	**−1.0271**	**−0.6251**	**−0.5546**	**0**
	${q_{1}}^{\mathrm{BES}}$	0.7628	0.1792	−0.1051	−0.6480	−2.8364	−0.4301	1.9628	0
	${q_{1}}^{\mathrm{PSO}}$	−2.4654	−0.8442	0.3242	−0.3084	1.5155	0.2270	1.8986	9.24 × 10^−8^
	${q_{1}}^{\mathrm{FA}}$	0.4883	1.2184	−0.0115	1.2445	−1.6829	−0.1029	0.3778	8.00 × 10^−3^
	${q_{1}}^{\mathrm{MTSA}}$	0.2956	0.4752	−0.5083	−0.1977	0.5249	1.2244	2.3713	1.35 × 10^−3^
	${q_{1}}^{\text{CPSO-AT}}$	−0.8107	0.7907	3.0000	−1.2719	3.0000	−0.5459	−2.1306	1.09 × 10^−2^
2	${q_{2}}^{\text{I-BES}}$	**−0.1098**	**−0.1094**	**0.4421**	**−0.9758**	**−0.3422**	**0.2167**	**0.1038**	**0**
	${q_{2}}^{\mathrm{BES}}$	0.4988	0.7104	1.0956	0.5850	0.1250	1.0202	0.9756	0
	${q_{2}}^{\mathrm{PSO}}$	−0.3010	0.7237	−2.6322	0.0127	2.9792	−2.2643	−1.7273	5.70 × 10^−8^
	${q_{2}}^{\mathrm{FA}}$	0.0607	1.2992	−0.7629	1.3140	3.0000	−0.9099	−2.3478	1.60 × 10^−3^
	${q_{2}}^{\mathrm{MTSA}}$	−0.3270	0.7632	2.3712	−1.0135	1.0739	1.0818	0.4062	2.31 × 10^−3^
	${q_{2}}^{\text{CPSO-AT}}$	−1.8167	−0.1816	3.0000	−0.5186	−2.0103	−0.6423	−3.0000	1.23 × 10^−2^
3	${q_{3}}^{\text{I-BES}}$	**0.0746**	**0.5572**	**1.0837**	**−0.1600**	**−0.7741**	**−1.4375**	**1.0498**	**0**
	${q_{3}}^{\mathrm{BES}}$	0.6574	0.6795	1.0394	0.1932	1.7909	0.8991	1.4087	0
	${q_{3}}^{\mathrm{PSO}}$	0.7845	0.4717	1.5159	0.3739	0.9472	−0.4860	2.9980	1.38 × 10^−7^
	${q_{3}}^{\mathrm{FA}}$	−0.7280	0.5903	3.0000	−0.7372	3.0000	−1.5695	3.0000	4.90 × 10^−3^
	${q_{3}}^{\mathrm{MTSA}}$	0.5397	0.0990	0.0750	−1.1789	2.4431	0.5989	0.1868	5.88 × 10^−4^
	${q_{3}}^{\text{CPSO-AT}}$	−0.6582	0.5778	3.0000	−1.1497	−3.0000	1.7054	−1.2714	2.04 × 10^−2^
4	${q_{4}}^{\text{I-BES}}$	**0.6667**	**0.8606**	**0.8363**	**0.6108**	**0.1613**	**−0.0287**	**0.5247**	**0**
	${q_{4}}^{\mathrm{BES}}$	−0.3470	0.1156	2.4528	−0.1455	2.2797	−2.3181	2.5180	0
	${q_{4}}^{\mathrm{PSO}}$	−1.7673	−0.1756	3.0000	−0.4854	−2.3160	−0.8351	1.9073	4.71 × 10^−8^
	${q_{4}}^{\mathrm{FA}}$	−1.6630	−1.1575	1.1886	−0.8765	3.0000	−0.6982	−2.9013	7.20 × 10^−3^
	${q_{4}}^{\mathrm{MTSA}}$	−1.8309	−1.0467	2.7539	0.7868	2.9363	−1.3449	1.9635	2.23 × 10^−3^
	${q_{4}}^{\text{CPSO-AT}}$	0.4150	0.2087	1.9278	0.2886	3.0000	1.4557	2.0795	3.42 × 10^−2^

**Table 12 biomimetics-09-00627-t012:** Comparison of statistical results of CEC2017 test functions.

Function	Indicator	I-BES	BES	PSO	FA	MTSA	CPSO-AT
F3	Best	**1.43×10^4^**	$5.47\;\times \;10^{4}$	$5.66\;\times \;10^{4}$	$1.62\;\times \;10^{5}$	$7.39\;\times \;10^{4}$	$1.70\;\times \;10^{5}$
	Mean	**2.78×10^4^**	$8.59\;\times \;10^{4}$	$1.24\;\times \;10^{5}$	$2.92\;\times \;10^{5}$	$1.32\;\times \;10^{5}$	$2.65\;\times \;10^{5}$
	Std	**8.05×10^3^**	$1.89\;\times \;10^{4}$	$5.96\;\times \;10^{4}$	$6.24\;\times \;10^{4}$	$3.22\;\times \;10^{4}$	$5.22\;\times \;10^{4}$
F8	Best	$9.57\;\times \;10^{2}$	$1.06\;\times \;10^{3}$	**9.36×10^2^**	$1.16\;\times \;10^{3}$	$1.02\;\times \;10^{3}$	$1.14\;\times \;10^{3}$
	Mean	**1.01×10^3^**	$1.14\;\times \;10^{3}$	$1.03\;\times \;10^{3}$	$1.28\;\times \;10^{3}$	$1.14\;\times \;10^{3}$	$1.23\;\times \;10^{3}$
	Std	**2.60×10^1^**	$3.69\;\times \;10^{1}$	$4.40\;\times \;10^{1}$	$6.01\;\times \;10^{1}$	$4.70\;\times \;10^{1}$	$2.88\;\times \;10^{1}$
F9	Best	$4.07\;\times \;10^{3}$	$6.67\;\times \;10^{3}$	**3.47×10^3^**	$1.23\;\times \;10^{4}$	$1.21\;\times \;10^{4}$	$1.59\;\times \;10^{4}$
	Mean	**5.50×10^3^**	$1.09\;\times \;10^{4}$	$6.81\;\times \;10^{3}$	$1.96\;\times \;10^{4}$	$1.73\;\times \;10^{4}$	$2.15\;\times \;10^{4}$
	Std	**6.42×10^2^**	$2.33\;\times \;10^{3}$	$2.44\;\times \;10^{3}$	$4.74\;\times \;10^{3}$	$2.63\;\times \;10^{3}$	$2.38\;\times \;10^{3}$
F14	Best	**1.63×10^3^**	$8.74\;\times \;10^{3}$	$1.87\;\times \;10^{3}$	$5.79\;\times \;10^{3}$	$4.91\;\times \;10^{3}$	$3.26\;\times \;10^{5}$
	Mean	**5.42×10^3^**	$2.17\;\times \;10^{5}$	$6.13\;\times \;10^{5}$	$3.11\;\times \;10^{6}$	$9.42\;\times \;10^{4}$	$1.91\;\times \;10^{6}$
	Std	**5.35×10^3^**	$3.40\;\times \;10^{5}$	$1.97\;\times \;10^{6}$	$4.95\;\times \;10^{6}$	$1.18\;\times \;10^{5}$	$1.56\;\times \;10^{6}$
F15	Best	**4.46×10^3^**	$9.06\;\times \;10^{3}$	$8.79\;\times \;10^{3}$	$3.57\;\times \;10^{5}$	$8.51\;\times \;10^{4}$	$5.37\;\times \;10^{7}$
	Mean	**2.57×10^4^**	$3.74\;\times \;10^{8}$	$1.62\;\times \;10^{5}$	$1.07\;\times \;10^{9}$	$3.00\;\times \;10^{5}$	$6.41\;\times \;10^{8}$
	Std	**2.53×10^4^**	$5.14\;\times \;10^{8}$	$1.43\;\times \;10^{5}$	$1.05\;\times \;10^{9}$	$1.20\;\times \;10^{5}$	$3.64\;\times \;10^{8}$
F18	Best	**1.77×10^4^**	$6.95\;\times \;10^{4}$	$3.92\;\times \;10^{4}$	$6.27\;\times \;10^{5}$	$8.90\;\times \;10^{4}$	$1.89\;\times \;10^{6}$
	Mean	**7.56×10^4^**	$4.17\;\times \;10^{6}$	$3.86\;\times \;10^{6}$	$2.92\;\times \;10^{7}$	$1.05\;\times \;10^{6}$	$1.74\;\times \;10^{7}$
	Std	**4.21×10^4^**	$5.82\;\times \;10^{6}$	$6.12\;\times \;10^{6}$	$3.98\;\times \;10^{7}$	$1.16\;\times \;10^{6}$	$1.33\;\times \;10^{7}$
F19	Best	**4.01×10^3^**	$2.18\;\times \;10^{6}$	$2.52\;\times \;10^{4}$	$3.20\;\times \;10^{7}$	$6.72\;\times \;10^{5}$	$1.12\;\times \;10^{8}$
	Mean	**4.87×10^5^**	$4.84\;\times \;10^{8}$	$1.96\;\times \;10^{7}$	$2.11\;\times \;10^{9}$	$3.05\;\times \;10^{6}$	$6.69\;\times \;10^{8}$
	Std	$1.92\;\times \;10^{6}$	$7.60\;\times \;10^{8}$	$5.50\;\times \;10^{7}$	$2.17\;\times \;10^{9}$	**1.35×10^6^**	$4.39\;\times \;10^{8}$
F27	Best	**3.20×10^3^**	$3.20\;\times \;10^{3}$	$3.28\;\times \;10^{3}$	$3.41\;\times \;10^{3}$	$3.35\;\times \;10^{3}$	$3.27\;\times \;10^{3}$
	Mean	**3.21×10^3^**	$3.25\;\times \;10^{3}$	$3.46\;\times \;10^{3}$	$3.63\;\times \;10^{3}$	$3.62\;\times \;10^{3}$	$3.31\;\times \;10^{3}$
	Std	$6.83\;\times \;10^{1}$	$2.47\;\times \;10^{2}$	$1.40\;\times \;10^{2}$	$1.61\;\times \;10^{2}$	$2.32\;\times \;10^{2}$	**2.51×10^1^**
F30	Best	**6.54×10^3^**	$6.69\;\times \;10^{7}$	$3.63\;\times \;10^{5}$	$1.51\;\times \;10^{8}$	$6.39\;\times \;10^{5}$	$5.07\;\times \;10^{7}$
	Mean	**1.16×10^6^**	$6.54\;\times \;10^{8}$	$6.78\;\times \;10^{7}$	$9.83\;\times \;10^{8}$	$4.28\;\times \;10^{6}$	$3.71\;\times \;10^{8}$
	Std	**2.17×10^6^**	$8.07\;\times \;10^{8}$	$2.52\;\times \;10^{8}$	$9.01\;\times \;10^{8}$	$3.57\;\times \;10^{6}$	$2.61\;\times \;10^{8}$

**Table 13 biomimetics-09-00627-t013:** Wilcoxon rank sum test result.

	I-BES vs. BES		I-BES vs. PSO		I-BES vs. FA		I-BES vs. MTSA		I-BES vs. CPSO-AT	
	* **p** * **Value**	**R**	* **p** * **Value**	**R**	* **p** * **Value**	**R**	* **p** * **Value**	**R**	* **p** * **Value**	**R**
F3	3.02×10^−11^	+	3.02×10^−11^	+	3.02×10^−11^	+	1.78×10^−10^	+	3.02×10^−11^	+
F8	1.33×10^−10^	+	8.19×10^−1^	+	3.02×10^−11^	+	7.38×10^−10^	+	3.02×10^−11^	+
F9	3.69×10^−11^	+	2.71×10^−1^	+	3.02×10^−11^	+	3.02×10^−11^	+	3.02×10^−11^	+
F14	9.53×10^−7^	+	3.96×10^−8^	+	2.61×10^−10^	+	1.87×10^−7^	+	3.02×10^−11^	+
F15	8.89×10^−10^	+	1.07×10^−7^	+	3.02×10^−11^	+	1.55×10^−9^	+	3.02×10^−11^	+
F18	3.34×10^−11^	+	7.39×10^−11^	+	3.02×10^−11^	+	6.70×10^−11^	+	3.02×10^−11^	+
F19	3.34×10^−11^	+	8.89×10^−10^	+	3.02×10^−11^	+	4.98×10^−11^	+	3.02×10^−11^	+
F27	8.56×10^−4^	+	6.53×10^−8^	+	1.10×10^−8^	+	1.56×10^−8^	+	8.84×10^−7^	+
F30	3.02×10^−11^	+	1.25×10^−7^	+	3.02×10^−11^	+	2.78×10^−7^	+	3.02×10^−11^	+

**Table 14 biomimetics-09-00627-t014:** Best results for tension/compression spring design problem.

Algorithm	Variable			Function Value
	$x_{1}\left(d\right)$	$x_{2}\left(D\right)$	$x_{3}\left(P\right)$	
I-BES	0.25000	0.25000	15.00000	**2.6952×10^6^**
BES	0.25000	0.25000	14.93521	2.8151×10^6^
PSO	0.25000	0.25000	15.00000	**2.6952×10^6^**
FA	0.25021	0.25000	15.00000	2.7875×10^6^
MTSA	0.25000	0.25000	15.00000	**2.6952×10^6^**
CPSO-AT	0.25001	0.25000	15.00000	2.6973×10^6^

**Table 15 biomimetics-09-00627-t015:** Statistical results of tension/compression spring design problem.

Algorithm	Best	Worst	Mean	Std
I-BES	2.6952×10^6^	4.6026×10^7^	8.1552×10^6^	1.1503×10^7^
BES	2.8151×10^6^	6.3855×10^7^	2.5655×10^7^	2.0781×10^7^
PSO	2.6952×10^6^	2.6953×10^6^	2.6952×10^6^	1.7202×10^2^
FA	2.7875×10^6^	6.8007×10^7^	1.1593×10^7^	1.5486×10^7^
MTSA	2.6952×10^6^	2.0086×10^7^	6.2766×10^6^	5.1281×10^6^
CPSO-AT	2.6973×10^6^	2.9661×10^6^	2.7765×10^6^	7.0300×10^4^

**Table 16 biomimetics-09-00627-t016:** Best results for pressure vessel design problem.

Algorithm	Variable				Function Value
	$x_{1}\left(T_{s}\right)$	$x_{2}\left(T_{h}\right)$	$x_{3}\left(R\right)$	$x_{4}\left(L\right)$	
I-BES	0.7782	0.3847	40.3223	200.0000	**5.8862×10^3^**
BES	1.0342	0.5131	53.5855	72.2317	6.4930×10^3^
PSO	0.8284	0.4534	42.884	167.1468	6.1251×10^3^
FA	0.7529	0.5573	38.8902	223.3197	6.4065×10^3^
MTSA	0.8012	0.3982	41.5697	184.4628	6.0799×10^3^
CPSO-AT	1.0473	1.5477	40.6484	200.0000	1.1426×10^4^

**Table 17 biomimetics-09-00627-t017:** Statistical results of pressure vessel design problem.

Algorithm	Best	Worst	Mean	Std
I-BES	5.8862×10^3^	1.1489×10^5^	2.0310×10^4^	2.4556×10^4^
BES	6.4930×10^3^	2.3109×10^5^	3.8668×10^4^	5.3125×10^4^
PSO	6.1251×10^3^	4.1947×10^5^	4.1409×10^4^	8.9159×10^4^
FA	6.4065×10^3^	7.5295×10^5^	1.5491×10^5^	1.9404×10^5^
MTSA	6.0799×10^3^	7.5439×10^4^	1.2482×10^4^	1.4948×10^4^
CPSO-AT	1.1426×10^4^	4.7605×10^4^	2.1284×10^4^	8.9094×10^3^

## Data Availability

The data generated during the current study are available from the corresponding author upon reasonable request. The data are available, but restrictions apply to the availability of these data, which were used under license for the current study and are not publicly available.

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
