# Peer review of "Improved Bald Eagle Search Optimization Algorithm for the Inverse Kinematics of Robotic Manipulators"

_biomimetics, 2024, doi:10.3390/biomimetics9100627_

Round 1

Reviewer 1 Report

Comments and Suggestions for Authors

1-) In the abstract, some statistical results must be reported.

2-) The grammatical or writing errors must be corrected in the paper.

3-) The introduction section was very weak; Some algorithms such as I-CPA, MS-TSA are also included in swarm algorithms and should be mentioned with their problems. Also this section should be further developed.

4-) The conclusion section was very weak; it will be improved and more efficient with the feature line. The conclusions must be improved. The authors must provide significant results.

5-) Literature on the application fields of metaheuristic algorithms is lacking. For example; https://doi.org/10.1016/j.egyr.2021.11.103, https://doi.org/10.3390/biomimetics8070540 doi's must strengthen the importance of metaheuristics by adding works on the application of metaheuristic algorithms to different problems. Not limited to these, strengthen the literature search to make the paper strong and effective.

6-) Share the code of the proposed method and the problem on publicly known platforms (like github or drive).

7-) The performance of the proposed algorithm must be tested on some benchmark problems such as CEC 2017. Compare the results of the proposed algorithm with the results of CEC2017 in “MS-TSA” and “I-CPA” algorithms. Likewise, a non-parametric test such as Wilcoxon or Friedman is necessary to compare the algorithms accurately.

8-) The authors must compare the proposed method with modern and some classical metaheuristics.

9-) It is not clearly stated or cited in the article by whom the BES algorithm was developed. Please clarify this situation.

Author Response

Response to reviewer 1

Dear Reviewer,

Thank you very much for your comments. In response to your suggestions, we have made changes to the corresponding parts of the article (highlighted in yellow). Our responses are provided below.

Comments 1: In the abstract, some statistical results must be reported.

Response 1: We added to the abstract the accuracy of the algorithm in solving the inverse kinematics problem as well as comparative results for comparison in the CEC2017 test set (page number: 1; line: 2). Below is the modified abstract:

The inverse kinematics of robotic manipulators involves determining an appropriate joint configuration to achieve a specified end-effector position. This problem is challenging because

the inverse kinematics of manipulators is highly nonlinear and complexly coupled. To address this

challenge, the bald eagle search optimization algorithm is introduced. This algorithm combines the

advantages of evolutionary and swarm techniques, making it more effective at solving nonlinear

problems and improving search efficiency. Due to the tendency of the algorithm to fall into local

optima, the Lévy flight strategy is introduced to enhance its performance. This strategy adopts a

heavy-tailed distribution to generate long-distance jumps, thereby preventing the algorithm from

getting trapped in local optima and enhancing its global search efficiency. The experiments first

evaluated the accuracy and robustness of the algorithm based on the inverse kinematics problem of

manipulators, achieving a solution accuracy of up to 10^(−18). Subsequently, the algorithms were tested and compared using the CEC2017 test functions, and the results showed that the improved algorithm outperformed the original and four popular metaheuristics in terms of accuracy, convergence speed, and stability. Specifically, it achieved over a 70% improvement in standard deviation and mean compared to the original algorithm in several test functions, demonstrating the effectiveness of the Lévy flight strategy in enhancing global search capability. Additionally, the practicality of the proposed algorithm was verified through two real engineering optimization problems.

Comments 2: The grammatical or writing errors must be corrected in the paper.

Response 2: We have read the article carefully and corrected the grammatical errors in the article. Here are the exact locations of the corrections:

(1)page number: 3; Figure1

(2)page number: 4; Eq. (2)

(3)page number: 7; line: 148

(4)page number: 8; line: 176

(5)page number: 10; line: 198

(6)page number: 11; line: 202

Comments 3: The introduction section was very weak; Some algorithms such as I-CPA, MS-TSA are also included in swarm algorithms and should be mentioned with their problems. Also this section should be further developed.

Response 3: We have made references to algorithms such as I-CPA as well as MS-TSA and the engineering problems corresponding to the algorithms in the introduction section (page number: 2; line: 32). We have simultaneously introduced the application of metaheuristic algorithms in engineering (page number: 2; line: 25) and relevant research on BES in the introduction section (page number: 2; line: 43).

Comments 4: The conclusion section was very weak; it will be improved and more efficient with the feature line. The conclusions must be improved. The authors must provide significant results.

Response 4: We revised the conclusion by using the simulation and experimental results from Section 5 as the main feature and provided meaningful results (page number: 27; line: 304). The following is the revised conclusion.

In this paper, an improved bald eagle search optimization method is proposed to enhance global search efficiency by introducing the Lévy flight strategy in the search phase of the original method, thereby preventing it from falling into local optima prematurely. Experimental results show that the convergence accuracy of I-BES can reach 10^(−18) when solving the inverse kinematics problem of manipulators. Moreover, I-BES outperforms BES and four other popular optimization methods on CEC2017 test functions in terms of search accuracy, convergence speed, and stability. Compared to BES, I-BES improves performance on most test functions by more than 70%. Finally, two additional case studies involving engineering optimization problems further validate that I-BES achieves better optimization results in real-world scenarios. Future research will focus on exploring its application to mobile and dual-arm manipulators.

Comments 5: Literature on the application fields of metaheuristic algorithms is lacking. For example; https://doi.org/10.1016/j.egyr.2021.11.103, https://doi.org/10.3390/biomimetics8070540 doi's must strengthen the importance of metaheuristics by adding works on the application of metaheuristic algorithms to different problems. Not limited to these, strengthen the literature search to make the paper strong and effective.

Response 5: We included the two provided references in the introduction to highlight the importance of meta-heuristic algorithms through their engineering applications (page number: 2; line: 25). We also expanded the literature review on swarm intelligence optimization algorithms (page number: 2; line: 32) and provided an overview of the relevant literature on BES (page number: 2; line: 43).

Comments 6: Share the code of the proposed method and the problem on publicly known platforms (like github or drive).

Response 6: We have uploaded our code to github. The code includes the comparison results of the proposed method (I-BES) with BES, PSO, FA, MS-TSA, and I-CPA on the CEC2017 test set. The research problem involves nine test functions (F3, F8, F9, F14, F15, F18, F19, F27, F30). Below is the link to the relevant code.

https://github.com/Autherparadox/CEC2017_test

Comments 7: The performance of the proposed algorithm must be tested on some benchmark problems such as CEC 2017. Compare the results of the proposed algorithm with the results of CEC2017 in “MS-TSA” and “I-CPA” algorithms. Likewise, a non-parametric test such as Wilcoxon or Friedman is necessary to compare the algorithms accurately.

Response 7: In Section 5.2 of the article, we added the comparison results of the proposed algorithm (I-BES) with MS-TSA and I-CPA on the CEC2017 test set (page number: 20-21; line: 270-275). Additionally, we included a wilcoxon non-parametric test to illustrate the differences between the proposed algorithm and other algorithms (page number: 22-23; line: 276-278).

Comments 8: The authors must compare the proposed method with modern and some classical metaheuristics.

Response 8: In Section 5, we compared the proposed method (I-BES) with classical meta-heuristic algorithms (PSO and FA) as well as modern meta-heuristic algorithms (BES, MS-TSA, and I-CPA). The comparisons include: (1) Section 5.1: the inverse kinematics problem of robotic manipulators (page number: 10-20; line: 189-270), (2) Section 5.2: tests on selected functions from the CEC2017 benchmark (page number: 20-23; line: 270-278), and (3) Section 5.3: tests on two other engineering problems (page number: 23-25; line: 279-290).

Comments 9: It is not clearly stated or cited in the article by whom the BES algorithm was developed. Please clarify this situation.

Response 9: We added citations for BES in the introduction (page number: 2; line: 43) and the section 3.1 (page number: 5; line: 91).

Reviewer 2 Report

Comments and Suggestions for Authors

The authors apply the Bald Eagle Search Optimization Algorithm for one of the classic optimization problems, and this approach does not have any novelty or interest for readers. I suggest to authors try to improve this algorithm and apply it to different engineering problems.

Author Response

Response to reviewer 2

Dear Reviewer,

Thank you very much for your comments. In response to your suggestions, we have made changes to the corresponding parts of the article (highlighted in yellow). Our responses are provided below.

Comments 1: The authors apply the Bald Eagle Search Optimization Algorithm for one of the classic optimization problems, and this approach does not have any novelty or interest for readers. I suggest to authors try to improve this algorithm and apply it to different engineering problems.

Response 1: To address the issue of the Bald Eagle Search (BES) optimization algorithm easily falling into local optima, this paper proposes an improved Bald Eagle Search optimization algorithm (I-BES). By incorporating the Lévy flight strategy into the search phase of BES, the algorithm can quickly escape from local optima and improve overall search efficiency (page number: 7; line: 131-139). Meanwhile, in the experimental section of the article, we validated the algorithm on two additional engineering design problems (the tension/compression spring design problem and the pressure vessel design problem) (page number: 23-25; line: 279-290).

Reviewer 3 Report

Comments and Suggestions for Authors

1. The literature review appears to be incomplete. For example, there is no mention of the SIN-COS Algorithm, Egle strategy, Golden Egle optimizer, Bald Egle search etc.

2. The formulation of the optimization problem solved by the algorithm is not provided

3. The method for forming the initial population is not described.

4. When finding new positions of population members, there is no check for the admissibility of solutions.

5. The conditions for ending the search process are not described.

6. The formula for the value x mean is not provided.

7. Formula (6) contains a variable x i +1, which may not exist.

8. The variables Rrand and rand are not described.

9. The results of the study of the efficiency of the algorithm on a generally accepted set of test functions are not provided.

Author Response

Response to reviewer 3

Dear Reviewer,

Thank you very much for your comments. In response to your suggestions, we have made changes to the corresponding parts of the article (highlighted in yellow). Our responses are provided below.

Comments 1: The literature review appears to be incomplete. For example, there is no mention of the SIN-COS Algorithm, Egle strategy, Golden Egle optimizer, Bald Egle search etc.

Response 1: We expanded the literature review and mentioned algorithms such as the SIN-COS Algorithm (page number: 2; line: 25), Eagle Strategy (page number: 2; line: 42), Golden Eagle Optimizer (page number: 2; line: 32), and Bald Eagle Search (page number: 2; line: 43).

Comments 2: The formulation of the optimization problem solved by the algorithm is not provided

Response 2: We have provided the formulation of the optimization problem solved by the algorithm (page number: 8; Eq. (12)).

Comments 3: The method for forming the initial population is not described.

Response 3: We have explained the generation method of the initial population in the article (page number: 9-10; line: 185-187).

Comments 4: When finding new positions of population members, there is no check for the admissibility of solutions.

Response 4: We have revised the pseudocode (page number: 9; Algorithm 1) and flowchart (page number: 10; Figure 6) in the article to ensure that the solution is validated when generating new positions for the population members.

Comments 5: The conditions for ending the search process are not described.

Response 5: We have revised the flowchart (page number: 10; Figure 6), and the modified version illustrates the termination conditions during the search process.

Comments 6: The formula for the value x mean is not provided.

Response 6: We have provided the formula for the value x mean in the article (page number: 6; line: 108).

Comments 7: Formula (6) contains a variable x i +1, which may not exist.

Response 7: To ensure xi+1 is meaningful, i takes values in the range [1, N-1] during the search phase. In the selection and swoop phases, i takes values in the range [1, N] (N represents the population size) (page number: 6; line: 122).

Comments 8: The variables Rrand and rand are not described.

Response 8: We have described the concepts of R (page number: 6; line: 120) and rand (page number: 6; line: 121) separately in the article.

Comments 9: The results of the study of the efficiency of the algorithm on a generally accepted set of test functions are not provided.

Response 9: In Section 5.2 of the article, we added the comparison results of the proposed algorithm (I-BES) with MS-TSA and I-CPA on the CEC2017 test set (page number: 20-21; line: 270-275). Additionally, we included a wilcoxon non-parametric test to illustrate the differences between the proposed algorithm and other algorithms (page number: 22-23; line: 276-278).

Round 2

Reviewer 1 Report

Comments and Suggestions for Authors

There are some serious errors in the article. These must be corrected. For example, it must be MTSA, not MS-TSA. It must be CPSO-AT (Improved Chaotic Particle Swarm Optimisation Algorithm), not I-CPA. You must correct this mistake in all texts, tables and figures. The MS-TSA (https://doi.org/10.1016/j.asoc.2024.112220) and I-CPA (https://doi.org/10.3390/biomimetics8080569) studies you have given are incorrectly stated in the conclusion section. The correct version of these studies is indicated in the doi's and you must present the correct version in the introduction. 

Other revisions have been made. 

Author Response

Response to reviewer 1

Dear Reviewer,

Thank you very much for your comments. In response to your suggestions, we have made changes to the corresponding parts of the article (highlighted in green). Our responses are provided below.

Comments 1: There are some serious errors in the article. These must be corrected. For example, it must be MTSA, not MS-TSA. It must be CPSO-AT (Improved Chaotic Particle Swarm Optimisation Algorithm), not I-CPA. You must correct this mistake in all texts, tables and figures. The MS-TSA (https://doi.org/10.1016/j.asoc.2024.112220) and I-CPA (https://doi.org/10.3390/biomimetics8080569) studies you have given are incorrectly stated in the conclusion section. The correct version of these studies is indicated in the doi's and you must present the correct version in the introduction.

Other revisions have been made.

Response 1: We have corrected the descriptions of the MTSA and CPSO-AT in Section 5 of the article, and here are the specific locations of the corrections:

Texts: (page number: 13; line: 216, 218), (page number: 14; line: 219, 220), (page number: 15; line: 222, 223), (page number: 17; line: 251), (page number: 18; line: 269), (page number: 20; line: 273), (page number: 21; line: 276), (page number: 24; line: 287), (page number: 25; line: 292, 298), (page number: 26; line: 302).

Tables: (page number: 11; Table 1), (page number: 14; Table 8), (page number: 19; Table 10, Table 11), (page number: 21; Table 12), (page number: 22; Table 13), (page number: 24; Table 14, Table 15), (page number: 25; Table 16, Table 17).

Figures: (page number: 13; Figure 8 to Figure 13), (page number: 15; Figure 14), (page number: 16; Figure 15), (page number: 22; Figure 19).

We have corrected the full names of the two algorithms that provide DOI (MS-TSA, I-CPA) in the introduction (page number: 2; line: 32).

We have also corrected the following errors in the article.

(page number: 1; line: 2), (page number: 109; line: 113), (page number: 8; line: 160), (page number: 14; line: 221), (page number: 18; line: 267, 268), (page number: 21; line: 277), (page number: 27; line: 306).

Best regards,

The authors

Reviewer 2 Report

Comments and Suggestions for Authors

acceptable 

Author Response

Response to reviewer 2

Dear Reviewer,

Thank you very much for recognizing the article.

Best regards,

The authors

Reviewer 3 Report

Comments and Suggestions for Authors

All major comments from the review have been corrected.

In lines 109,113 and equation (12), the variable i should be written in italics.

Author Response

Response to reviewer 3

Dear Reviewer,

Thank you very much for your comments. In response to your suggestions, we have made changes to the corresponding parts of the article (highlighted in green). Our responses are provided below.

Comments 1: All major comments from the review have been corrected. In lines 109,113 and equation (12), the variable i should be written in italics.

Response 1: We have modified the variable i at the corresponding position in the article (page number: 6; line: 109, 113), (page number: 8; line: 160).

Best regards,

The authors
